# Local synthesis of dynein cofactors matches retrograde transport to acutely changing demands

Joseph M. Villarin[1], Ethan P. McCurdy[2], José C. Martínez[1] & Ulrich Hengst[3,4]

Cytoplasmic dynein mediates retrograde transport in axons, but it is unknown how its transport characteristics are regulated to meet acutely changing demands. We find that stimulus-induced retrograde transport of different cargos requires the local synthesis of different dynein cofactors. Nerve growth factor (NGF)-induced transport of large vesicles requires local synthesis of Lis1, while smaller signalling endosomes require both Lis1 and p150$^{Glued}$. Lis1 synthesis is also triggered by NGF withdrawal and required for the transport of a death signal. Association of Lis1 transcripts with the microtubule plus-end tracking protein APC is required for their translation in response to NGF stimulation but not for their axonal recruitment and translation upon NGF withdrawal. These studies reveal a critical role for local synthesis of dynein cofactors for the transport of specific cargos and identify association with RNA-binding proteins as a mechanism to establish functionally distinct pools of a single transcript species in axons.

[1] Medical Scientist Training Program, College of Physicians and Surgeons, Columbia University, New York, New York 10032, USA. [2] Integrated Program in Cellular, Molecular and Biomedical Studies, College of Physicians and Surgeons, Columbia University, New York, New York 10032, USA. [3] The Taub Institute for Research on Alzheimer's Disease and the Aging Brain, College of Physicians and Surgeons, Columbia University, New York, New York 10032, USA. [4] Department of Pathology and Cell Biology, College of Physicians and Surgeons, Columbia University, New York, New York 10032, USA. Correspondence and requests for materials should be addressed to U.H. (email: uh2112@cumc.columbia.edu).

Many cellular functions rely on the ordered transport of macromolecules, proteins and organelles. Most intracellular transport is an active process mediated by motor protein complexes that move their cargos along components of the cytoskeleton: myosins transport cargos along actin filaments, while microtubule-based transport is facilitated by two families of motors, the plus-end directed kinesins and minus-end directed dynein[1]. In contrast to the great variety of myosins and kinesins, there is only a single cytoplasmic dynein, which is complemented by an array of regulatory proteins to fulfil different functions[2]. These multifunctional proteins bind either to the non-catalytic domains of dynein or directly to its force-generating heavy chain, thereby changing the characteristics of the dynein motor. For example, Lis1 (gene: PAFAH1B1) induces a persistent force-producing state in microtubules-attached, moving dynein by acting as a clutch linking the ATPase and microtubules-binding domains[3,4]. It maintains the microtubule-bound state of dynein[5], and is required for moving large vesicles through a constraint environment with high drag forces such as kinked axons[6]. Together with NudE, it enhances the sustained force produced by the dynein motor in a load induced manner[7]. In addition, axonal Lis1 has been described as an initiation factor for dynein-mediated transport[8]. Another dynein regulator, dynactin, is a large complex of eleven protein subunits, with p150$^{Glued}$, encoded by DCTN1, being the largest and most important. Among several described functions, dynactin acts by increasing the processivity of the dynein motor[9] and facilitates its binding to different cargos[10]. Importantly, dynactin and Lis1 competitively bind the same domain of dynein[11], suggesting a mutually exclusive regulation of dynein by these adaptors. Thus, cofactors or adaptor proteins such as dynactin or Lis1 regulate dynein-dependent transport, but it remains unknown how their association with dynein is controlled in a spatially precise and temporally acute manner in response to extracellular signals. This question is especially relevant in axons, where essentially all microtubules are unidirectionally oriented with their plus-ends facing the cellular periphery[12], and dynein is anterogradely transported as cargo by kinesins[13,14]. A potential solution to this question is the on-demand, local synthesis of dynein cofactors within distal axons and growth cones.

Intra-axonal protein synthesis is crucial for axon development[15–18], maintenance[19], synapse formation[20] and axo-somatic communication[21], as well as for axonal regeneration[22] and neurodegeneration[23]. From these studies, a picture emerges in which local protein synthesis provides short-lived and spatially precise bursts of locally translated proteins, to react to extracellular cues, injurious insults or other changes in an axon's environment[24]. Therefore, it is especially interesting that messenger RNAs (mRNAs) coding for dynein regulators, including Lis1 and p150$^{Glued}$, have repeatedly been found in axons both in the central and peripheral nervous system[23,25,26].

Here we asked whether local synthesis of dynein regulators was a mechanism to acutely match the intra-axonal retrograde transport capabilities to changes in demand, as, for example, in response to changes in extracellular trophic support. We report that axonal synthesis of Lis1 and p150$^{Glued}$ is required for the adjustment of retrograde transport to acutely changing neurotrophin signalling in the periphery of neurons.

## Results

**NGF-induced changes in retrograde transport require translation.** To investigate whether changes in intra-axonal transport required local protein synthesis, rat embryonic dorsal root ganglion (DRG) were grown in tripartite microfluidic chambers that allow fluidic isolation of distal axons from neuronal cell bodies and dendrites,

providing an experimental platform to study localized signalling events in axons (Fig. 1a)[17,27]. We first investigated the requirement of axonal protein synthesis for axonal retrograde transport in DRG neurons kept at a low nerve growth factor (NGF) concentration of 5 ng ml$^{-1}$ that is sufficient to support their survival. Application of protein synthesis inhibitors, anisomycin or emetine, did not change the proportion of retrogradely moving LysoTracker-positive particles, which in axons are mainly late endosomes and autophagosomes (Fig. 1b)[28]. Because transport in the 5 ng ml$^{-1}$ NGF condition was protein synthesis independent, we decided to use it as the baseline NGF concentration and to investigate whether NGF withdrawal (0 ng ml$^{-1}$) or stimulation (100 ng ml$^{-1}$) changed the transport of LysoTracker-positive particles in a protein synthesis-dependent manner. Upon NGF withdrawal or stimulation, retrograde transport of LysoTracker-positive vesicles was significantly increased with a corresponding decrease in the proportion of stationary vesicles (Fig. 1c,d; Supplementary Movies 1–5), while the percentages of anterogradely or bidirectionally moving particles were not significantly changed with either NGF concentration. Inhibition of protein synthesis completely abolished the increases in retrograde transport upon NGF stimulation and, surprisingly, upon NGF withdrawal. Activation of protein synthesis had before only been described in response to NGF stimulation[17,18,21,29,30], but not depletion. To investigate our finding that increased retrograde transport of LysoTracker-positive vesicles in NGF-deprived axons was sensitive to protein synthesis inhibition, we performed immunofluorescence against a marker of active protein synthesis, the phosphorylated form of 4EBP1 (Fig. 1e). The ratio of phosphorylated 4EBP1 was significantly increased within distal axons upon 10 min of NGF stimulation as well was withdrawal. Inhibition of mTOR with locally applied rapamycin completely abolished these changes. To directly visualize local protein synthesis in response to changes in NGF concentrations, we performed puromycylation assays. Puromycin is a transfer RNA mimetic that gets incorporated into nascent polypeptides and can be detected with specific antibodies[31]. NGF withdrawal and stimulation significantly increased the number of puromycylation events in axons in a protein synthesis inhibitor-sensitive manner, confirming that local protein synthesis is activated by both NGF stimulation and depletion (Fig. 1f). Puromycylation in the cell bodies was not affected by changes in NGF concentration or the addition of protein synthesis inhibitors in the axon compartment further proving the local nature of the NGF-induced changes in protein synthesis (Fig. 1g). These results establish that while constitutive, unstimulated retrograde transport does not require local protein synthesis, rapid increases in dynein-dependent transport of LysoTracker-positive particles in response to either NGF stimulation or withdrawal are mediated by axonally produced proteins.

**NGF stimulation or withdrawal affect Lis1 and p150$^{Glued}$.** mRNAs coding for regulators of cytoplasmic dynein have been found in several axonal transcriptomes (Fig. 2a)[23,25,26]. To investigate which proteins might be locally synthesized in response to NGF stimulation or withdrawal, we focused on Lis1 and p150$^{Glued}$. To directly visualize their mRNAs, Pafah1b1 and Dctn1 within axons of DRG neurons, we used fluorescence in situ hybridization (FISH). Both mRNAs were readily detectable in a punctate pattern in axons and with significantly higher intensity than the one obtained with a Gfp control probe (Fig. 2b). Using quantitative immunofluorescence, we found that the axonal abundance of Lis1 protein was significantly increased upon both NGF stimulation and withdrawal for 10 min (Fig. 2c), while in contrast p150$^{Glued}$ levels were elevated only in response to NGF stimulation (Fig. 2d). The levels of each protein were not changed

by pre-incubation with protein synthesis inhibitors under baseline conditions, but the increases in abundance upon NGF stimulation (for both Lis1 and p150$^{Glued}$) or withdrawal (Lis1 only) were abolished by the application of anisomycin or emetine to the axonal compartment. Together, these data indicate that the axonally localized transcripts of Lis1 and p150$^{Glued}$ might be translated in response to changes in NGF signalling.

**NGF signalling controls local Lis1 and p150$^{Glued}$ synthesis.** To directly test whether changes in axonal NGF signalling trigger the local synthesis of Lis1 and p150$^{Glued}$, we selectively transfected axons with siRNAs targeting their mRNAs *Pafah1b1* or *Dctn1*, respectively, or with a non-targeting control siRNA. We had validated the siRNAs by transfecting them into rat C6 glioma cells and immunoblotting whole-cell lysates for Lis1 and p150$^{Glued}$.

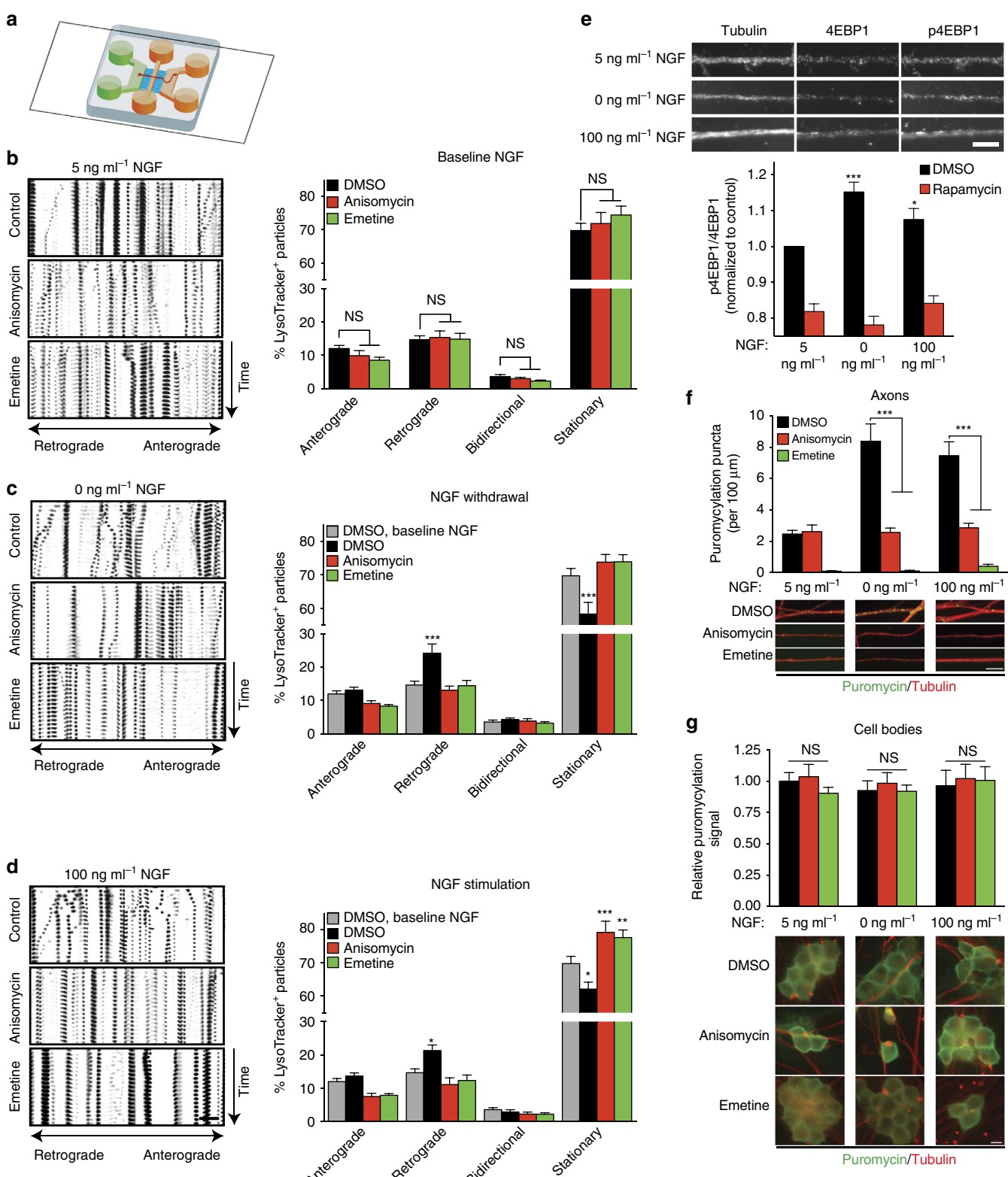

Two siRNAs against each transcript were tested individually and showed consistent phenotypes; the siRNAs were most efficacious when used together (Supplementary Fig. 1). Previously, we have demonstrated that the RNA interference (RNAi) pathway is functional in developing axons[32], and that it is possible to selectively knockdown an mRNA in axons by local siRNA transfection without causing a decrease of the transcript's abundance in cell bodies[17,18,23,33]. We confirmed that the effects of the local siRNA transfections were indeed restricted to axons by quantitative immunofluorescence against Lis1 and p150$^{Glued}$ on cell bodies whose axons had been transfected with siRNAs. No decrease of either protein was detectable in the neuronal soma (Fig. 3a,b). In the siRNA-transfected axons, the protein levels were not significantly reduced at our baseline NGF concentration, again indicating that the mRNAs are not locally translated under this condition (Fig. 3c,d). Conversely, the significant increases in Lis1 abundance in response to NGF stimulation or withdrawal were completely abolished by local siRNA application, as was the increase in p150$^{Glued}$ levels in NGF-stimulated axons. Together, these results demonstrate that both transcripts can be locally translated in axons, but the intra-axonal synthesis of these two dynein cofactors is differentially regulated by changes in NGF signalling.

**Lis1 synthesis is required for NGF-induced vesicle transport.** To determine whether the local synthesis of Lis1 and p150$^{Glued}$ in response to changes in NGF concentration impacted retrograde transport in axons, we incubated axons with LysoTracker and scored the motility of labelled vesicles as before. Axonal knockdown of *Pafah1b1* or *Dctn1* did not significantly affect retrograde transport in the baseline condition (Fig. 4a), in line with our finding that neither protein is locally synthesized under baseline conditions. Axon-specific knockdown of *Pafah1b1* abolished the significant increase in the proportion of retrogradely moving vesicles caused by NGF deprivation (Fig. 4b; Supplementary Movie 6), and it caused a reduction of retrogradely moving LysoTracker-positive particles below the baseline levels in the NGF-stimulated condition (Fig. 4c; Supplementary Movie 7). In contrast, knockdown of axonal *Dctn1* mRNA did not affect the movement of LysoTracker-positive vesicles upon either NGF stimulation of withdrawal. Together, these results demonstrate that locally synthesized Lis1 is required for induced retrograde movement of these LysoTracker-labelled cargos, but p150$^{Glued}$ is not. This observation is reminiscent of the finding that, globally, high load retrograde transport requires Lis1 (ref. 6).

**NGF–endosome transport requires Lis1 and p150$^{Glued}$ synthesis.** To investigate whether the requirement for local synthesis of

dynein cofactor varied between different cargos, we next visualized the retrograde transport of NGF-signalling endosomes[34]. Upon binding of NGF to its receptor TrkA, the receptor–ligand complex is internalized, and the resulting endosome is transported with downstream effector complexes to the soma by a dynein–dynactin complex[35]. Mouse 2.5S NGF was linked to red fluorescent quantum dots (QDs)[36], selectively applied to axons (100 ng ml$^{-1}$), and movement of QD-labelled NGF-signalling endosomes was measured by live-cell microscopy (Fig. 4d). The proportion of retrogradely moving particles seen under naive and control siRNA conditions (~27%) was consistent with previous studies[37,38]. Axon-specific knockdown of *Pafah1b1* or *Dctn1* significantly reduced the retrograde movement of QD-positive particles and increased the proportion of stationary particles. Together, these results establish that NGF stimulation triggers local synthesis of Lis1 and p150$^{Glued}$, and that the stimulated transport of different cargos requires the local synthesis of different regulator proteins.

**Transport of an axonal death signal requires Lis1 synthesis.** According to the signalling endosomes hypothesis, transport of NGF bound to activated tyrosine receptor kinases in endosomes from axons to the cell body is required for the survival of neurons dependent upon target-derived neurotrophic support[35,39], while another model proposes that NGF acts by suppressing a retrograde apoptotic signal, and that retrograde transport of NGF-signalling endosomes is not required for survival[40]. Because of the observed reduction in retrogradely moving, QD-labelled NGF-signalling endosomes upon axon-specific knockdown of *Pafah1b1* or *Dctn1* mRNAs, we next tested whether survival of the DRG neurons was impaired as well. NGF was withheld from both compartments or selectively applied (100 ng ml$^{-1}$) to the axonal compartment. To quench any residual NGF activity in the deprivation conditions, a neutralizing anti-NGF antibody was added. Contrary to what the signalling endosomes hypothesis would predict but in line with a suppressive effect of NGF on an axonal apoptotic signal in starved axons, in the NGF-replete condition axonal knockdown of either *Pafah1b1* or *Dctn1* did not induce apoptosis, as assessed by TUNEL-positive nuclei, nor did it reduce the number of living neurons stained by calcein acetoxymethyl (AM) (Fig. 5a,b). Moreover, in the NGF-starved condition, knockdown of *Pafah1b1* completely prevented the induction of cell death by NGF deprivation. Knockdown of *Dctn1* in the NGF-starved condition did not impact cell death, consistent with our finding that NGF withdrawal does not activate p150$^{Glued}$ synthesis.

Together, these results demonstrate that, although inhibition of local Lis1 and p150$^{Glued}$ synthesis greatly reduced retrograde transport of NGF-signalling endosomes, their local production is

**Figure 1 | Local protein synthesis mediates NGF-regulated changes in axonal transport.** (**a**) Representation of a microfluidic chamber used to isolate axons. DRG neurons are seeded in the cell body compartment (green), and the axons extend through two microgroove barriers (blue) into the axonal compartments (orange). All axon-specific treatments were applied to both axonal compartments, and analyses were performed in the distal most compartment. (**b–d**) DRG neurons were cultured in microfluidic chambers for 3 DIV, at which point the NGF concentration in the axonal chamber was changed to 5 ng ml$^{-1}$ for 24 h. On DIV 4, axons were pretreated with protein synthesis inhibitors (anisomycin and emetine) or vehicle (dimethylsulphoxide, DMSO) for 2 h before application of medium containing the inhibitors or DMSO and either 5 ng ml$^{-1}$ NGF (**b**), no NGF (**c**), or 100 ng ml$^{-1}$ NGF (**d**) and LysoTracker Green for 15 min. Live-imaging time-lapse series of axonal fields were acquired, with images being taken every 13 s for 4 min. Kymographs of representative 100-μm-long axonal segments are shown. Scale bar, 10 μm. LysoTracker-positive particles with diameters ≥1 μm were scored as anterograde, retrograde, bidirectional or stationary. Percentage point differences to baseline condition are plotted. Data represent the means ± s.e.m. of nine fields per conditions (n = 3 biological replicates). *P ≤ 0.05; **P ≤ 0.01; ***P ≤ 0.001. One-way ANOVA with Bonferroni's multiple comparison test. (**e**) DRG neurons were cultured as in **b**. After 10 min of different NGF treatments, axonal levels of 4EBP1 and p-4EBP1 were determined by immunofluorescence. Scale bar, 5 μm. Data represent the means ± s.e.m. of 15 fields per conditions (n = 3 biological replicates). *P ≤ 0.05; ***P ≤ 0.001. Two-way ANOVA with Dunnett's multiple comparison test. (**f,g**) DRG neurons were cultured and axons were treated with NGF and inhibitors as in **b**. Puromycin was added to all compartments of the chambers during the NGF treatment period. *P ≤ 0.01; ***P ≤ 0.001. Two-way ANOVA with Bonferroni's multiple comparison test. Scale bars, 10 μm. NS, not significant.

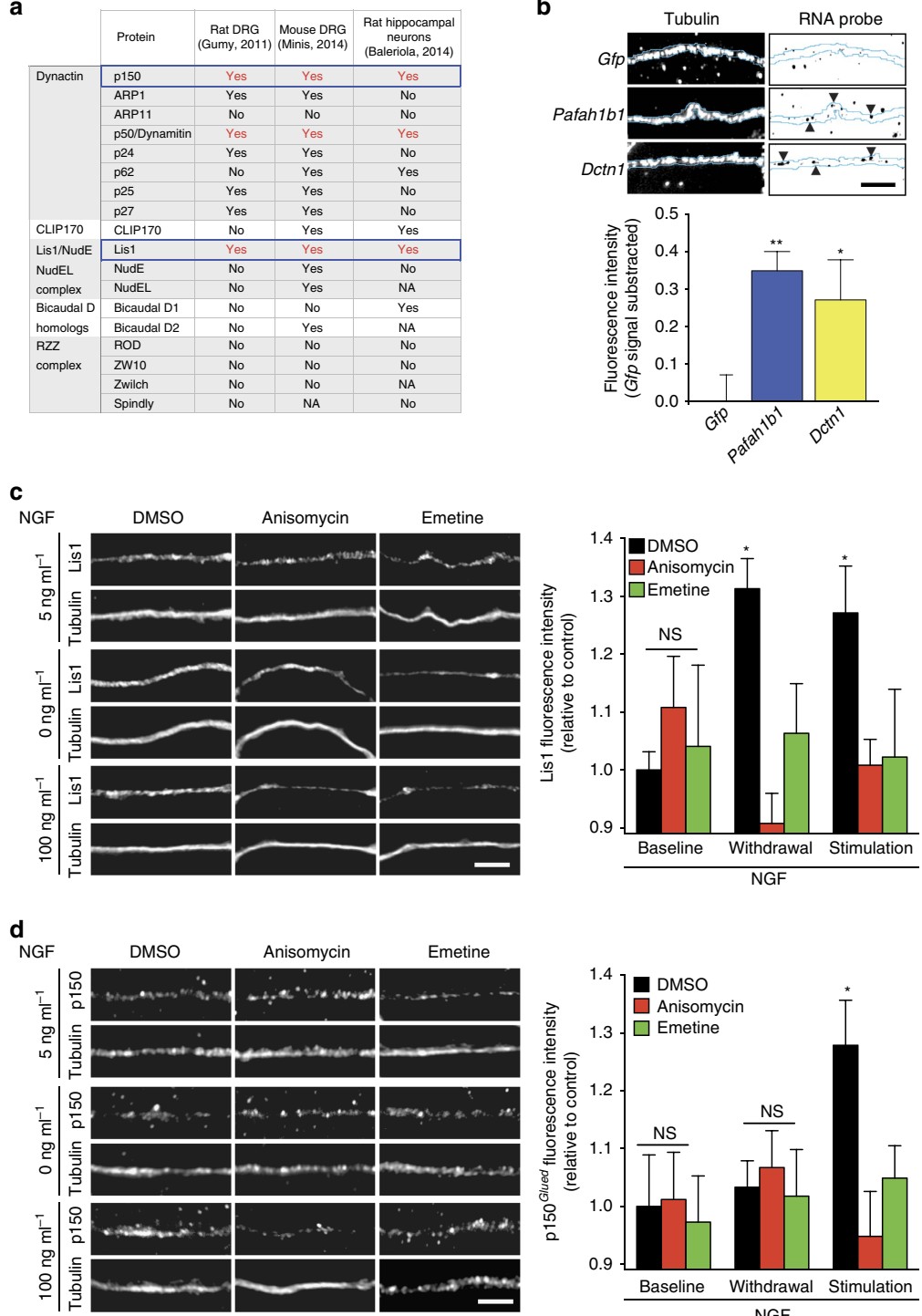

**Figure 2 | NGF signalling differentially regulates Lis1 and p150$^{Glued}$ levels in axons.** DRG neurons were cultured and treated as in Fig. 1. (**a**) Transcripts coding for dynein regulators have been found in transcriptomes derived from embryonic rat DRG axons using microarray, embryonic mouse DRG using RNAseq and embryonic rat hippocampal axons using RNAseq. Transcripts found in all three studies are highlighted in red, and Lis1 and p150$^{Glued}$ are outlined in blue. (**b**) *Pafah1b1* and *Dctn1* levels were measured by quantitative FISH in axons kept for 12 h at the baseline NGF level (5 ng ml$^{-1}$). Background fluorescence was determined using a *Gfp* probe and subtracted. Means ± s.e.m. of 15 optical fields per condition (*n* = 3 biological replicates). *$P \leq 0.05$; **$P \leq 0.01$. Kruskal–Wallis test with Dunn's multiple comparison test. (**c**) Axons were pretreated with protein synthesis inhibitors (anisomycin and emetine) or vehicle, followed by exposure to different concentrations of NGF (0, 5 or 100 ng ml$^{-1}$) for 10 min. Axonal Lis1 levels were measured by quantitative immunofluorescence. Means ± s.e.m. of 15–20 optical fields per conditions (*n* = 3–4 biological replicates). *$P \leq 0.05$. Two-way ANOVA with Dunnett's multiple comparison test. (**d**) Neurons were cultured and treated as in **b**. Axonal p150$^{Glued}$ levels were measured by quantitative immunofluorescence. Means ± s.e.m. of 15 optical fields per conditions (*n* = 3 biological replicates). *$P \leq 0.05$. Two-way ANOVA with Dunnett's multiple comparison test. Scale bars, 5 μm. NS, not significant.

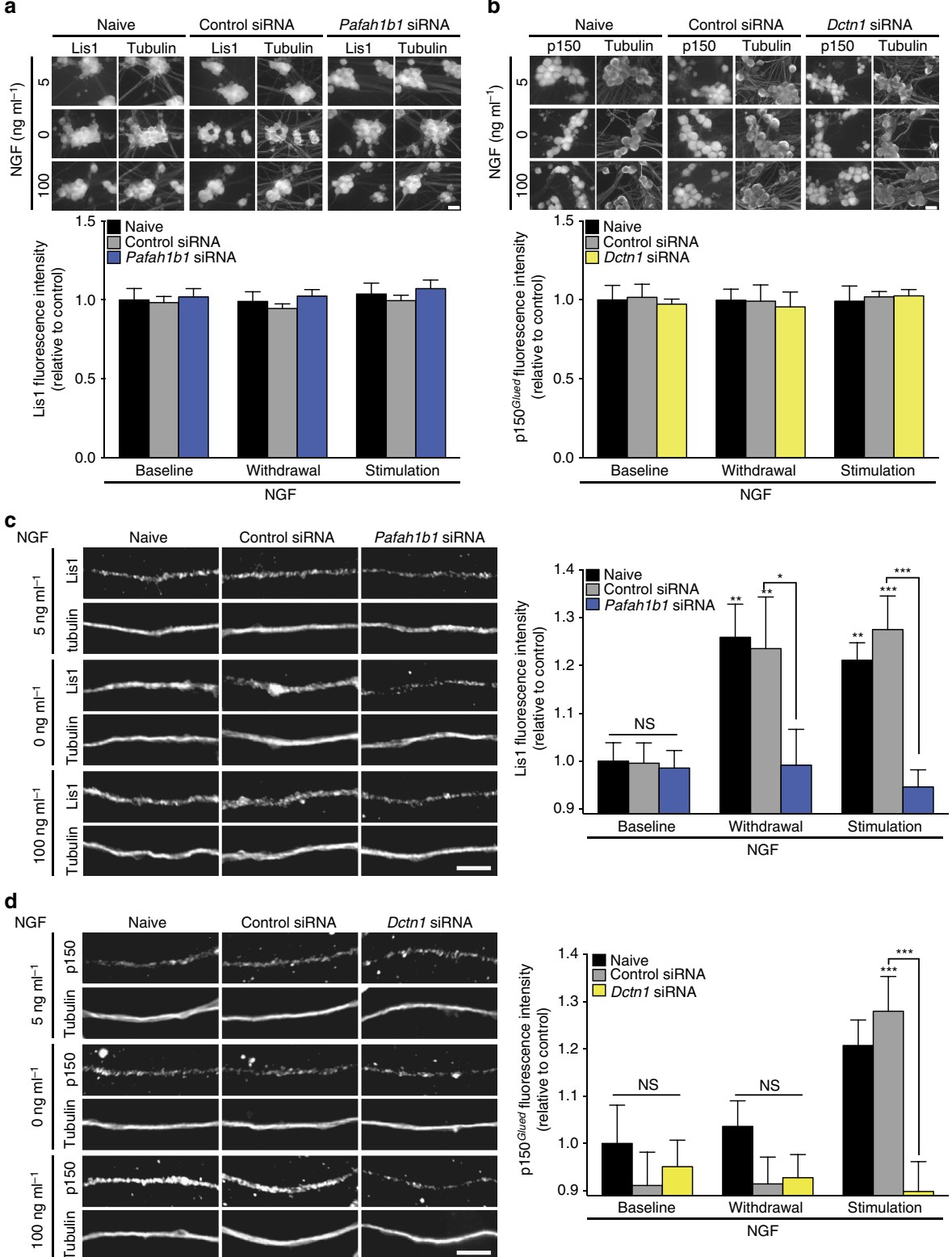

**Figure 3 | NGF induces local synthesis of Lis1 and p150$^{Glued}$.** DRG neurons were cultured in microfluidic chambers. On DIV 3, the NGF concentration in the axonal chamber was changed to 5 ng ml$^{-1}$, and axons were selectively transfected with a non-targeting control siRNA or siRNAs targeting *Pafah1b1* or *Dctn1*. (**a,b**) Twenty-four hours after transfection, axons were treated with 0, 5 or 100 ng ml$^{-1}$ NGF for 10 min, and Lis1 (**a**) and p150$^{Glued}$ (**b**) levels in the cell bodies were determined by immunofluorescence. Means ± s.e.m. of 15 optical fields per conditions (*n* = 3 biological replicates). No significant changes. Two-way ANOVA with Dunnett's multiple comparison test. Scale bars, 20 μm. (**c,d**) Neurons were cultured and treated as before, and axonal Lis1 (**c**) and p150$^{Glued}$ (**d**) levels were determined by immunofluorescence. Scale bars, 5 μm. Means ± s.e.m. of 20–75 optical fields per conditions (*n* = 4–15 biological replicates). *$P \leq 0.05$; **$P \leq 0.01$; ***$P \leq 0.001$. Two-way ANOVA with Dunnett's multiple comparison test. See also Supplementary Fig. 1. NS, not significant.

not required for NGF-dependent survival. Rather, local Lis1 synthesis is necessary for the retrograde transport of a pro-apoptotic signal of unknown identity that is generated in NGF-deprived axons[40,41]. To further characterize this retrograde death signal ,we first focused on protein kinases that have been implicated in apoptotic cell death in neurons. Whole-cell treatment with inhibitors of mixed lineage kinases or p38 MAP kinase has previously been shown to prevent neuronal apoptosis induced by neurotrophin deprivation[42,43], but application of these inhibitors to axons alone did not interfere with induction of

apoptosis in NGF-deprived DRGs, suggesting that these kinases act centrally rather than in the periphery (Fig. 5c). GSK3β had been proposed as a carrier of the axonally generated apoptotic signal[40]. Application of two GSK3 inhibitors, LiCl or SB216763 (ref. 44), selectively to axons had no effect on cell death under NGF-replete conditions, but completely prevented the induction of apoptosis with NGF deprivation (Fig. 5c). Together, these results indicate that the death signal, whose transport requires local Lis1 production, involves active GSK3β.

**NGF signalling regulates axonal *Pafah1b1* and *Dctn1* levels.** In regenerating DRG axons, neurotrophins regulate the abundance of specific mRNAs through anterograde recruitment from the cell body[45]. To investigate whether changes in neurotrophin signalling not only differentially regulate the translation of axonally localized *Pafah1b1* and *Dctn1* mRNAs but also their abundance, we performed quantitative FISH on axons selectively transfected with siRNAs and treated with different NGF concentrations. As before, the effect of the siRNAs was restricted to axons as neither mRNA's abundance in the neuronal cell bodies was changed upon axonal siRNA transfection (Fig. 6a,b). Quantification of the axonal FISH signals revealed that neither mRNA was recruited in response to stimulation with NGF, but that NGF deprivation caused a significant increase in Lis1 transcript levels (Fig. 6c,d). Similar results were obtained for FISH against the transcripts of NudE and its paralogue NudEL, two proteins can form a trimeric complex with dynein and Lis1 (Supplementary Fig. 2)[46], indicating that NGF might co-regulate mRNAs of proteins that frequently function in a complex. The FISH signal was specific for the targeted mRNAs as transfection of axons with siRNAs targeting either transcript reduced the FISH signal to background levels. The results of the FISH experiments were confirmed by quantitative real-time PCR with reverse transcription (RT–PCR) performed on RNA collected from axonal compartments (Fig. 6e). The siRNAs failed to reduce the levels of their target transcripts under NGF conditions that do not trigger the translation of these mRNAs (baseline for both mRNAs; NGF withdrawal for *Dctn1*). This effect is likely due to the tight packaging in RNA granules of silenced mRNAs in axons[47] rendering them inaccessible for the RNAi machinery, an effect we had observed previously[23]. Together, these results demonstrate that the intra-axonal expression of the dynein

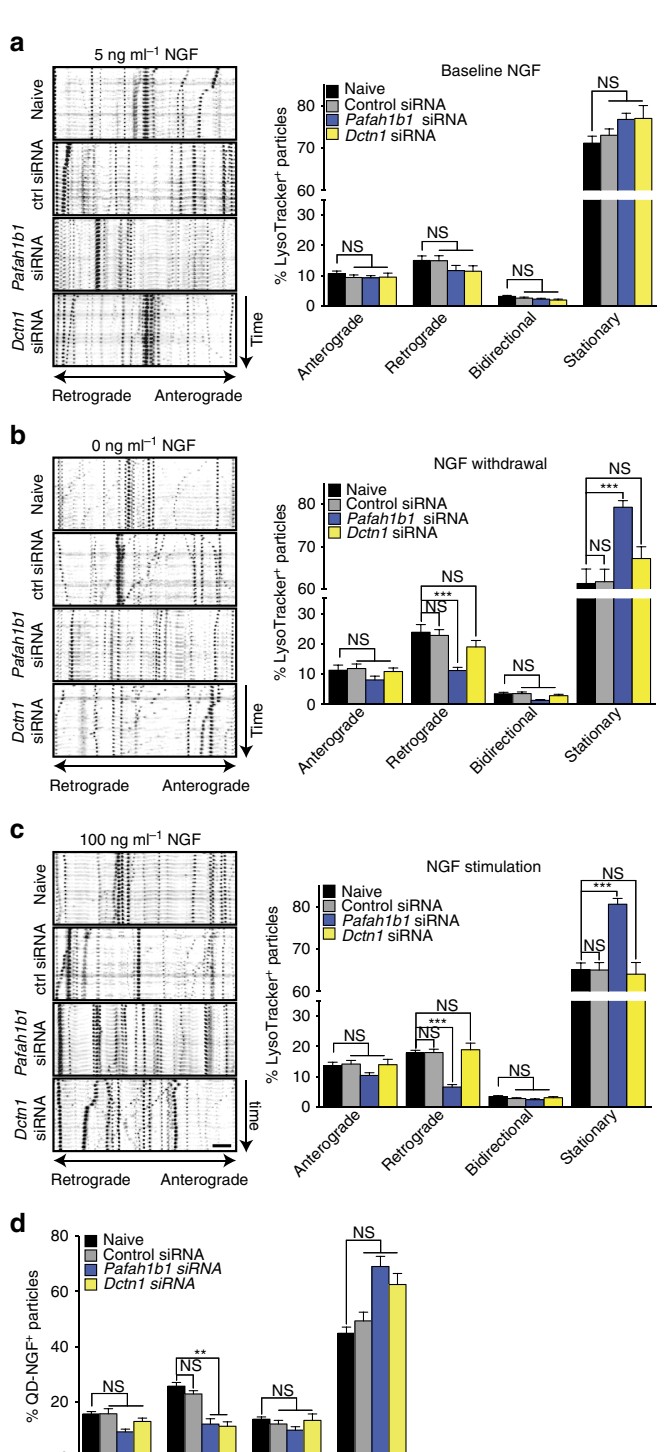

**Figure 4 | NGF-induced changes in axonal trafficking require local synthesis of Lis1 or p150*Glued*.** DRG neurons were cultured in microfluidic chambers. On DIV 3, the NGF concentration in the axonal chamber was changed to 5 ng ml$^{-1}$, and axons were selectively transfected with a non-targeting control siRNA or siRNAs targeting *Pafah1b1* or *Dctn1*. (**a–c**) After 24 h, fresh medium was added to the axonal chamber containing 5 ng ml$^{-1}$ NGF, no NGF or 100 ng ml$^{-1}$ NGF together with LysoTracker Green for 15 min. Live-imaging time-lapse series of axonal fields were acquired, with images being taken every 13 s for 4 min. Kymographs of representative 100-μm-long axonal segments are shown. Scale bar, 10 μm. LysoTracker-positive particles with diameters ≥1 μm were scored as anterograde, retrograde, bidirectional or stationary. Means ± s.e.m. of 12–18 optical fields per conditions (n = 3–6 biological replicates). **P ≤ 0.01; ***P ≤ 0.001. One-way ANOVA with Bonferroni's multiple comparisons test. (**d**) On DIV 4, axons were treated with 100 ng ml$^{-1}$ QD-NGF for 15 min and live imaged as above. QD-labelled particles <1-μm diameter were scored as anterograde, retrograde, bidirectional or stationary. Means ± s.e.m. of nine optical fields per conditions (n = 3 biological replicates). **P ≤ 0.01; ***P ≤ 0.001. Kruskal–Wallis test with Dunn's multiple comparison test. NS, not significant.

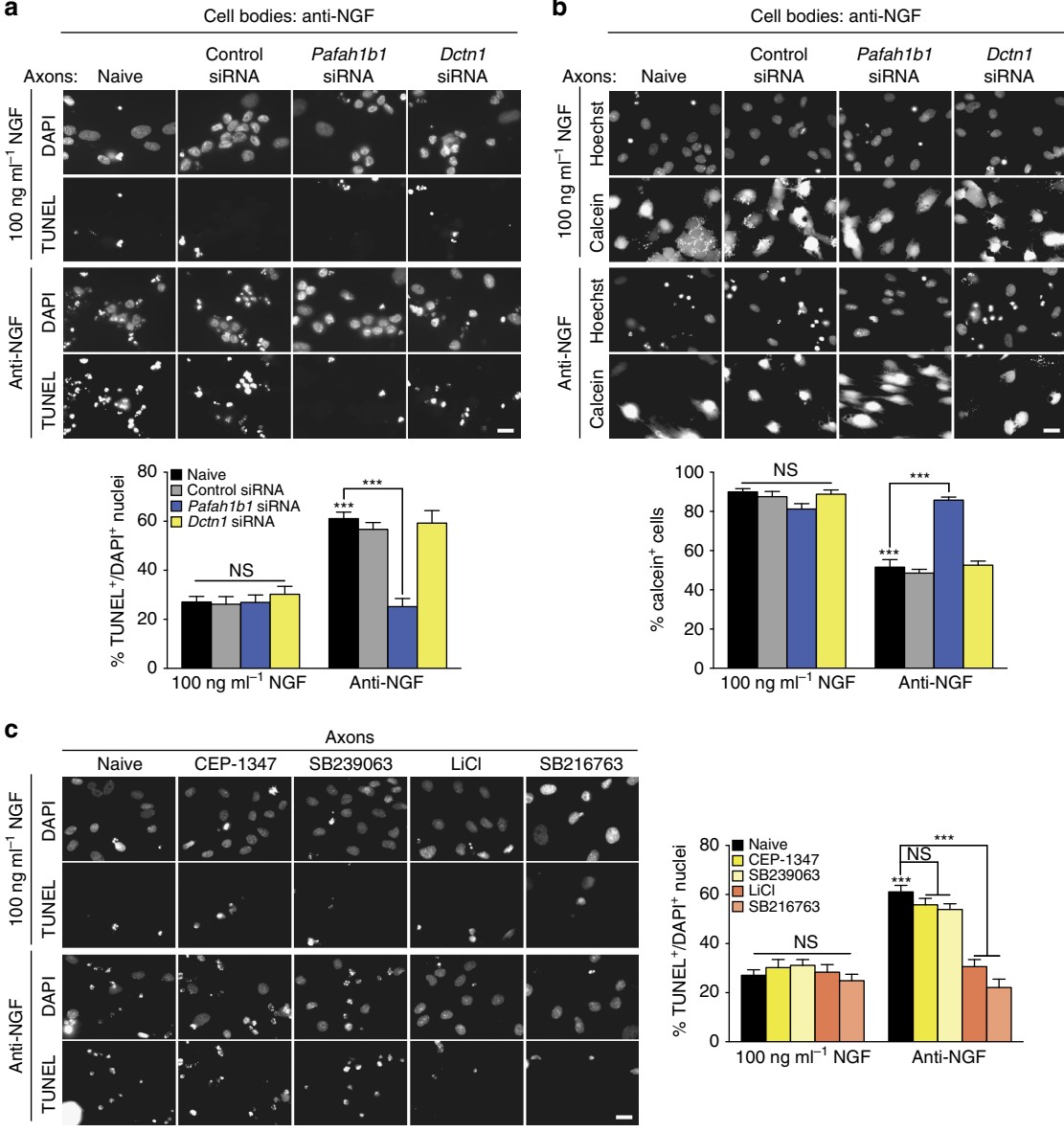

**Figure 5 | Pro-apoptotic signalling from NGF-deprived axons requires axonally synthesized Lis1 and active GSK3β.** (**a**) DRG neurons were cultured and transfected with siRNAs as in Fig. 4. On DIV 4, the medium in the somatic compartment was changed to NGF-free medium containing NGF-neutralizing antibody, and axonal compartments were changed to 100 ng ml$^{-1}$ NGF or NGF-free medium with NGF-neutralizing antibody plus vehicle for 24 h. Cell death was assessed by TUNEL assay. Means ± s.e.m. of 15–25 optical fields per conditions ($n=3$–5 biological replicates). ***$P\leq0.001$. Two-way ANOVA with Dunnett's multiple comparison test. (**b**) Neurons were cultured and treated as in **a**. Survival was assessed by calcein AM staining. Means ± s.e.m. of 15 optical fields per conditions ($n=3$ biological replicates). ***$P\leq0.001$. Two-way ANOVA with Dunnett's multiple comparison test. (**c**) DRG neurons were cultured as in Fig. 4. On DIV 4, the medium in the somatic compartment was changed to NGF-free medium containing NGF-neutralizing antibody, and the medium in the axonal chamber was changed to 100 ng ml$^{-1}$ NGF or NGF-free medium with NGF-neutralizing antibody plus the mixed lineage kinase inhibitor, CEP-1347, the p38 MAP kinase inhibitor, SB239063, or the GSK3β inhibitors, LiCl or SB216763, or vehicle for 24 h. Cell death was assessed by TUNEL assay. Means ± s.e.m. of 15–25 optical fields per conditions ($n=3$–5 biological replicates). ***$P\leq0.001$. Two-way ANOVA with Dunnett's multiple comparison test. Scale bars, 20 μm. NS, not significant.

cofactors Lis1 and p150$^{Glued}$ is differentially regulated both translationally and through recruitment of their mRNAs.

**APC-binding sorts *Pafah1b1* into functionally distinct pools.** The finding that the one mRNA species, *Pafah1b1*, is locally translated in response to both NGF stimulation and withdrawal, and, further, is recruited into axons only upon NGF deprivation but not stimulation, strongly suggested that distinct regulatory mechanisms exist that control *Pafah1b1* localization and

translation under different signalling conditions. Recently, *Pafah1b1* has been found to be part of the adenomatous polyposis coli (APC) interactome[48]. APC is a microtubules plus-end tracking protein, also referred to as +TIP[49], and by binding a specific subset of mRNAs, APC might provide a platform for the local synthesis of dynein regulators, including Lis1, at the distal end of axonal microtubules. Thus, we wondered whether association with APC was required for *Pafah1b1* regulation in axons. To address this question, we used a locked nucleic acid (LNA) oligomer designed to interfere with *Pafah1b1*–APC

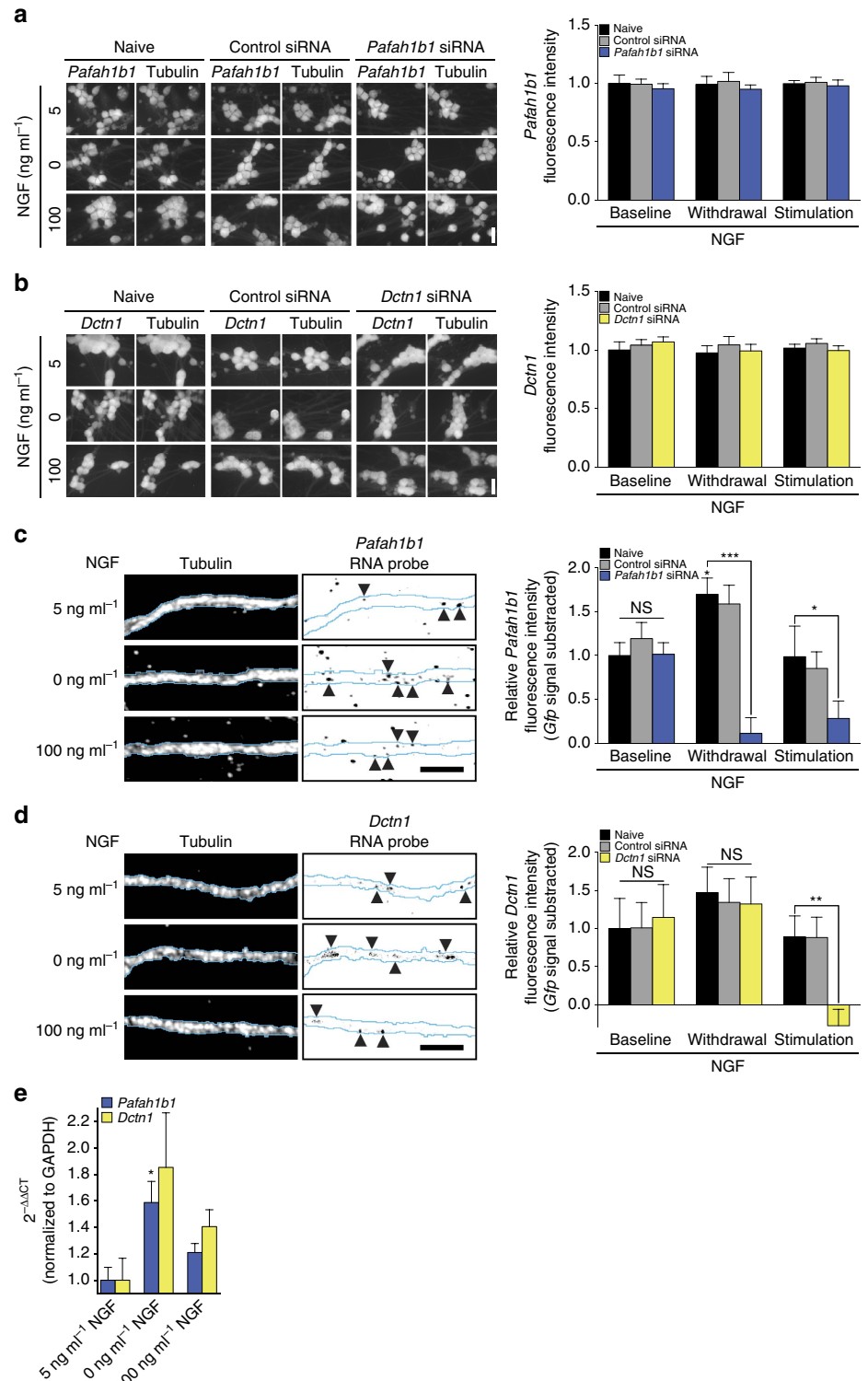

**Figure 6 | NGF signalling regulates axonal transcript levels of dynein regulators.** (**a–d**) DRG neurons were cultured in microfluidic chambers for 3 DIV, at which time the NGF concentration in the axonal chamber was changed to 5 ng ml$^{-1}$, and axons were selectively transfected with a non-targeting control siRNA or siRNAs targeting *Pafah1b1* (**a,c**) or *Dctn1* (**b,d**). Twenty-four hours after the transfection, the NGF concentration in the axonal chamber was adjusted to 0, 5 or 100 ng ml$^{-1}$ NGF for 12 h, and cell body *Pafah1b1* (**a**) or *Dctn1* (**b**) or axonal *Pafah1b1* (**c**) or *Dctn1* (**d**) mRNA levels were determined by FISH. Means ± s.e.m. of 15–25 optical fields per condition ($n = 3$–5 biological replicates). *$P \leq 0.05$; **$P \leq 0.01$; ***$P \leq 0.001$. Two-way ANOVA. Scale bars, 20 μm (**a,b**); 5 μm (**c,d**). (**e**) Neurons were cultured and axons treated with NGF in microfluidic chambers as before. Axonal RNAs were collected after the 12 h NGF treatment, and *Pafah1b1* and *Dctn1* levels were determined by quantitative real-time RT–PCR. Relative quantification with *Gapdh* as reference was done using the $2^{-\Delta\Delta CT}$ method. The means of the 5 ng ml$^{-1}$ NGF conditions for *Pafah1b1* and *Dctn1* were defined as 1.0. Means ± s.e.m. of 3–5 biological replicates. *$P \leq 0.05$. Kruskal–Wallis test with Dunn's multiple comparison test. NS, not significant.

association by binding the putative APC-binding site, a CUGU motif in the 3′-untranslated region (UTR) of *Pafah1b1* (ref. 48). To determine which of the several CUGU motifs in the 3′-UTR of *Pafah1b1* to target, reads from the APC-CLIP study[48] were collapse and quality-filtered[50] and mapped to the mouse genome (mm10). Mapped reads were analysed for cluster enrichment using PIPE-CLIP[51]. Three clusters in the 3′-UTR of *Pafah1b1* were found to be significantly enriched in APC-binding, but only

two of these clusters had a significant fold change compared with a control mRNA-seq data set. We chose the cluster with the lowest *P* value ($1.41 \times 10^{-11}$), which also contained a CUGU motif. A second LNA, binding *Pafah1b1* 13 bases upstream of the CUGU LNA, was used as a control (Fig. 7a). To confirm whether the CUGU LNA was able to interfere with APC–Pafah1b1 interaction, we transfected the LNAs in dissociated DRG and performed anti-APC RNA immunoprecipitation. *Pafah1b1* was

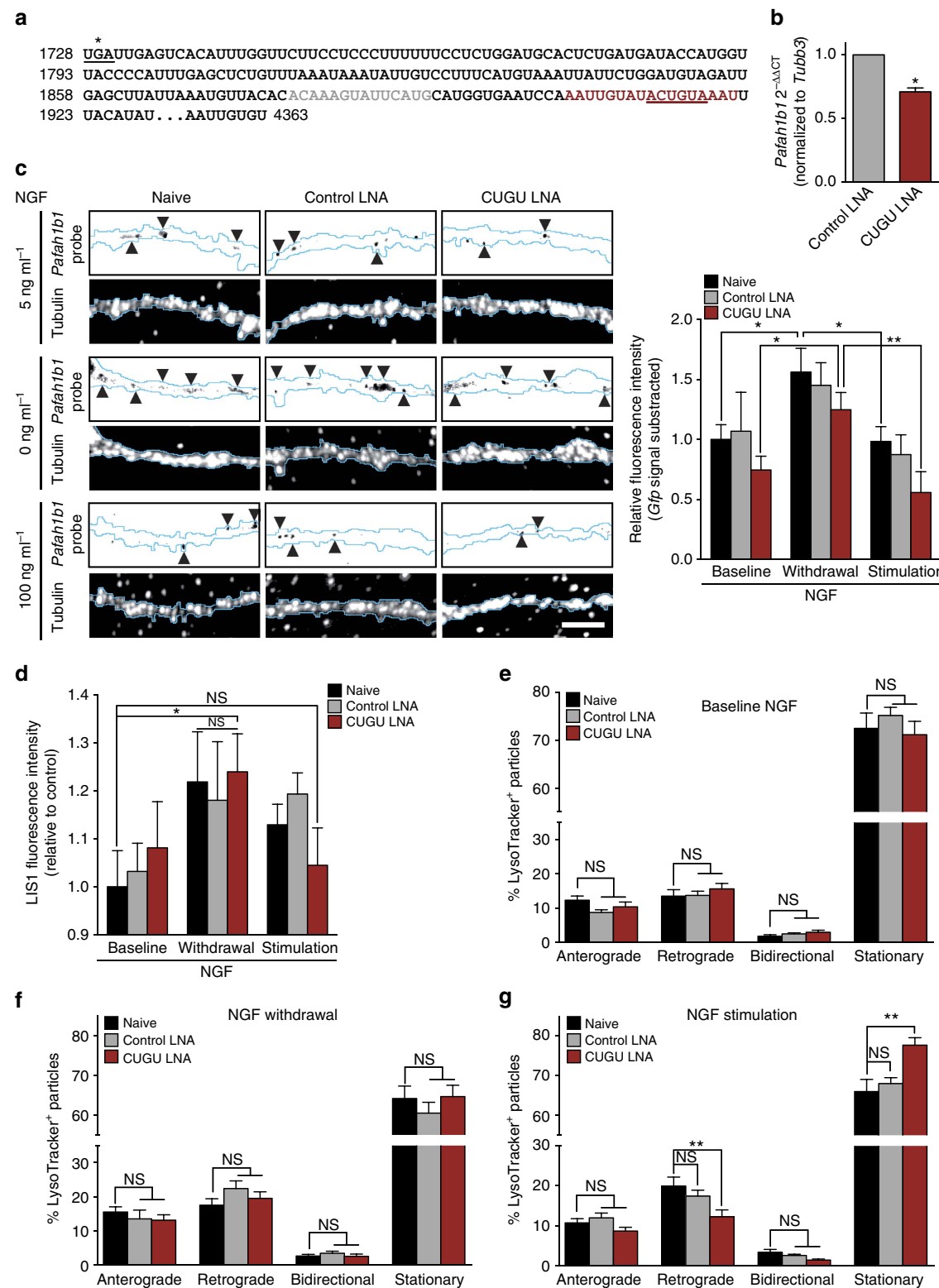

detectable in the precipitate and its abundance reduced in RIPs from CUGU LNA-transfected DRGs (Fig. 7b). Next, the LNAs were transfected in the cell body compartment, and mRNA levels in axons were determined by quantitative FISH 12 h after different NGF treatments as before (Fig. 7c). The control LNA had no discernible effect compared with naive axons (two-way analysis of variance (ANOVA) $P = 0.7585$), while transfection with the CUGU LNA caused reduced *Pafah1b1* levels in all three conditions ($P = 0.008$). The differences in *Pafah1b1* abundance at 5 and 0 ng ml$^{-1}$ or 0 and 100 ng ml$^{-1}$ NGF were significant in both naive and CUGU LNA-transfected axons and extremely similar (5 and 0 ng ml$^{-1}$: 0.56 versus 0.50; 0 and 100 ng ml$^{-1}$: 0.58 versus 0.68). These results indicate that APC association is required for the axonal localization of a fixed amount of *Pafah1b1*, but that the recruitment of additional *Pafah1b1* into axons in response to NGF withdrawal is APC-independent.

To determine whether Lis1 protein levels in axons were similarly affected by interference with *Pafah1b1*–APC binding, we transfected DRGs with LNAs as before and performed quantitative immunofluorescence against Lis1 (Fig. 7d). No significant change in Lis1 abundance was detected at 5 ng ml$^{-1}$ NGF, again confirming that it is not locally synthesized under this condition. The increase in axonal Lis1 abundance in CUGU LNA-transfected DRGs at 0 ng ml$^{-1}$ was indistinguishable from naive and control LNA neurons, while in NGF-stimulated axons transfection with the CUGU LNA prevented the increase in Lis1 levels.

Finally, we investigated the requirement of *Pafah1b1*–APC association for stimulation-induced retrograde transport of LysoTracker-positive cargos. Transfection of the CUGU LNA had no effect on transport at the baseline NGF condition or upon NGF withdrawal, but completely prevented the increase in retrograde transport triggered by NGF stimulation (Fig. 7e–g).

Together, the results from the LNA experiments reveal the existence of two distinct modes of *Pafah1b1* localization in axons: one that is constitutively active and APC-dependent, and other that is APC-independent and particularly responsive to NGF deprivation (Supplementary Fig. 3). The transcripts that are localized in the APC-dependent mode are translated in response to stimulation with NGF, while the APC-independent pool is locally translated with NGF withdrawal and is boosted by recruitment into NGF-starved axons. Thus, the two diametrically opposed triggers of axonal Lis1 synthesis, NGF withdrawal and stimulation, act on two separate pools of *Pafah1b1* mRNA that each are solely responsible for the increase in local Lis1 levels under either condition.

## Discussion

Association with various accessory proteins allows cytoplasmic dynein to fulfil a multitude of functions in cells and to transport a wide variety of different cargos. Here we provide evidence that in distal axons, stimulus-induced changes in dynein-dependent transport are regulated through local synthesis of Lis1 and p150$^{Glued}$. The unidirectional nature of microtubules in axons poses special challenges for the acute regulation and initiation of dynein-dependent transport, and local translation of its cofactors can solve this problem. As the unidirectional orientation of microtubules is not unique to axons but occurs also in distal dendrites or during neuronal cell migration, it is possible that this mechanism is utilized in these circumstances as well. Meanwhile, local synthesis of motor complex proteins might not be restricted to Lis1 and p150$^{Glued}$. p150$^{Glued}$ is only one of several subunits of dynactin. The transcript for another subunit, p50, is also consistently found in transcriptomes, while the localization of transcripts coding for other subunits is less clear. It remains unknown whether the entire dynactin complex can be locally synthesized or whether it locally assembles upon on-demand synthesis of p150$^{Glued}$ and potentially p50.

Changes in intra-axonal transport have long been recognized as hallmarks of many neurological and neurodegenerative disorders[52,53]. In addition, alterations in neurotrophin signalling have been implicated in neurodegenerative and psychiatric disorders[54]. Therefore, it will be important to investigate whether the processes uncovered here are disrupted in disease and whether they might present novel targets for therapies. For example, our previous finding that soluble oligomeric A$\beta_{1-42}$ upregulates protein synthesis in axons of mature hippocampal neurons[23] indicates the possibility that local translation has an impact on dynein-dependent transport in Alzheimer's disease brain.

It is worth noting that axonal production is not merely supplementary to global synthesis of these dynein cofactors. Neuron-wide knockdown of Lis1 expression reduces retrograde stimulation-independent transport of LysoTracker-positive vesicles in axons[55], while we found that axon-specific knockdown of Lis1 or p150$^{Glued}$ prevented only induced changes of transport. Why is induced but not constitutively active transport dependent on local translation? A possible explanation might be that in response to stimulation, previously inactive dynein motor complexes get activated and coupled to their cargos. We found that Lis1 synthesis in response to NGF stimulation requires the association of *Pafah1b1* with APC. As a + TIP, APC is well situated to mediate the activation of dynein through local production of regulatory proteins. In fact, the

---

**Figure 7 | Association with APC separates axonally localized Lis1 transcripts into two functionally distinct pools.** (**a**) Partial sequence of the 3′-UTR of rat *Pafah1b1* starting at the stop codon (*). The binding regions of the CUGU and control LNAs are indicated in maroon and grey, respectively. The CUGU element is underlined. (**b**) Dissociated DRG were transfected with control and CUGU LNA, and 24 h later, APC RNA immunoprecipitation was performed. *Pafah1b1* was quantified by RT–PCR. $2^{-\Delta\Delta CT}$ values are reported relative to *Tubb3* (positive control, binds APC but is not targeted by the LNAs). *Gfp* was included as a control (no reads detected). Means ± s.e.m. ($n = 2$ biological replicates with two technical replicates each). *$P \leq 0.05$. t-test. (**c**) DRG neurons were cultured in microfluidic chambers. On DIV 3, the NGF concentration in the axonal chamber was changed to 5 ng ml$^{-1}$, and cell bodies were selectively transfected with the control or CUGU LNAs. Twenty-four hours after transfection, axons were treated with 0, 5 or 100 ng ml$^{-1}$ NGF for 12 h, and axonal *Pafah1b1* mRNA levels were determined by FISH. Background fluorescence was determined using a *Gfp* probe and subtracted. Means ± s.e.m. of 15 optical fields per condition ($n = 3$ biological replicates). *$P \leq 0.05$. Two-way ANOVA with Fisher's least significant difference test. Scale bar, 5 μm. (**d**) DRG neurons were cultured and transfected as in **a**. Twenty-four hours after transfection, axons were treated with 0, 5 or 100 ng ml$^{-1}$ NGF for 10 min, and axonal Lis1 protein levels were measured by quantitative immunofluorescence. Means ± s.e.m. of 20–30 optical fields per conditions ($n = 4$–6 biological replicates). *$P \leq 0.05$. Two-way ANOVA with Fisher's LSD test. (**e**–**g**) DRG neurons were cultured and transfected as in **a**. Twenty-four hours after transfection, transport of LysoTracker-positive particles was observed in axons at baseline NGF (**e**), without NGF (**f**) or stimulated with NGF (**g**). Live-imaging time-lapse series of axonal fields were acquired, with images being taken every 13 s for 4 min. LysoTracker-positive particles with diameters $\geq 1$ μm were scored as anterograde, retrograde, bidirectional or stationary. Means ± s.e.m. of nine optical fields per conditions ($n = 3$ biological replicates). **$P \leq 0.01$. One-way ANOVA with Bonferroni's multiple comparisons test. NS, not significant.

recruitment of dynactin by +TIPs has been found to be required for the initiation of retrograde axonal transport of various cargos[56]. In *Aspergillus nidulans*, the Lis1 homolog has also been described as an initiation factor for dynein-mediated transport that is absent from and is unnecessary for dynein–cargo complexes once they are in motion[8]. Local synthesis of Lis1 or p150^Glued at very precise loci in axons or growth cones could, therefore, be a tuning or initiation mechanism for dynein-based transport.

In essentially all instances, intra-axonal protein synthesis has been seen to be stimulus-dependent. Our finding that NGF deprivation triggers axonal Lis1 synthesis within 10 min leads to the question as to how withdrawal of a ligand can be a stimulus for translation. The extremely short time needed to induce translation rules out that it might be a consequence of neuronal degeneration caused by the lack of trophic support. Rather, translation appears to be triggered by a signalling pathway that is active in the absence of NGF and suppressed by NGF-TrkA binding. For example, TrkA has been proposed to act as a dependence receptor that triggers cell death in the absence of its ligand[57]. Our study provides additional support for the dependence receptor hypothesis and, further, an experimental paradigm in which to dissect the underlying cell intrinsic death pathway downstream of TrkA.

Our finding that association with APC establishes distinct pools of axonally localized *Pafah1b1* mRNA that differ as to whether they are translated in response to NGF stimulation or withdrawal provides mechanistic insight into the differential regulation of axonally localized mRNAs. APC localizes mRNAs to microtubule plus-ends, and spatially orchestrates protein synthesis in axons and growth cones[48]. It remains unknown how many of these translational hubs exist in axons. The finding that the netrin receptor DCC binds components of the protein synthesis machinery and regulates local translation[58] suggests that APC is not unique. In the fungus *Ustilago maydis*, polysomes are actively transported on the surface of early endosomes and likely translationally active there[59]. If the same occurs in neurons, it would suggest that cargos might be able to hitch a ride on dynein by locally synthesizing adaptor proteins on their surface.

In conclusion, through these studies, we provide a mechanistic explanation for how a unidirectional motor can be tuned to fulfil changing transport needs far away from the cell soma, and we further reveal that transcripts of the same gene exist in axons in functionally distinct pools based on their association with translational hubs.

## Methods

**Compartmentalized DRG culture.** All reagents were from ThermoFisher Scientific (Waltham, MA) unless otherwise noted. To apply NGF, inhibitors or siRNAs specifically to distal axons and growth cones without affecting the cell bodies, DRG neurons were prepared from Sprague-Dawley embryonic day 15 rat embryos of both sexes. All work involving animals was performed in accordance with the National Institutes of Health guidelines for the care and use of laboratory animals, and was approved by the Institutional Animal Care and Use Committee of Columbia University. Embryonic rat DRGs were grown in tripartite microfluidic chambers composed of three compartments (width of middle compartment: 500 μm; side compartments: 1,500 μm) connected by two microgroove barriers (microgroove length: 500 μm, width: 10 μm, height: 3 μm)[17,27]. Microfluidic chambers were produced according to published protocols[18,60]. The microfluidic chambers were coated with 100 μg ml$^{-1}$ poly-L-lysine (Trevigen, Gaithersburg, MD). The plating medium (Neurobasal, 1× B27, 2 mM glutamate, 20 μM 5'-fluorodeoxyuridine, 100 ng ml$^{-1}$ NGF) was completely exchanged for 5 ng ml$^{-1}$ NGF in both axonal compartments after 48 h. siRNA transfection in the axonal compartments or LNA transfection in the somatic compartment was performed on DIV 3, and all experiments were performed on DIV 4. Whenever stated, the axonal compartments only were treated with anisomycin (1 μM, Sigma-Aldrich), emetine (2 μM, EMD Millipore), rapamycin (10 nM, Sigma-Aldrich), CEP-1347 (0.5 μM, Sigma-Aldrich), SB239063 (1 μM, Sigma-Aldrich), lithium chloride (15 mM) or SB216763 (10 μM, Sigma-Aldrich).

**Live imaging of axonal cargos.** Axonal transport of various cargos was visualized using an Axio Observer.Z1 inverted microscope equipped with an AxioCam MRm Rev. 3 camera (Zeiss, Thornwood, NY). To assay the effects of acute changes in NGF signalling on axonal trafficking, 50 nM LysoTracker Green DND-26 was added to axons when the axonal medium was changed to the experimental NGF concentrations (0, 5 or 100 ng ml$^{-1}$), 15 min before the start of imaging. For imaging transport of NGF-containing endosomes, QD-NGF was prepared by mixing mouse NGF 2.5S-Biotin (Alomone Labs, Jerusalem) and Qdot 585 Streptavidin Conjugate in a 1:1.2 molar ratio, and incubating them together at 4 °C with continuous inversion for 24 h. QD-NGF was diluted to 100 ng ml$^{-1}$ and added to axons with a medium change 15 min before imaging. During imaging, neurons were kept in a CO$_2$- and humidity-controlled incubation chamber maintained at 37 °C. Images were acquired every 13 s over a total 4-min time period, with three fields of axons imaged per replicate. For motility analysis, LysoTracker-positive particles were scored only if they were ≥1 μm in diameter, thereby allowing the identification and tracing of individual particles, whereas the much scarcer and smaller QD-NGF particles were included only if they were ≤0.5 μm in diameter. Particles were scored as stationary, anterograde, retrograde or bidirectional according the following definitions: stationary if they travelled a distance <1 μm during the observation period; anterograde or retrograde if they displaced >3 μm in one direction; and bidirectional if they travelled >3 μm in both directions.

**Puromycylation assay.** To visualized locally synthesized proteins, puromycin (1.8 μM) was added to axons or cell bodies for 10 min. Incorporation of puromycin into nascent polypeptide chains was determined by quantitative immunofluorescence and quantified as the numbers of puromycin-positive puncta in axons or relative intensity of the puromycin immunofluorescence signal.

**Fluorescence *in situ* hybridization.** Antisense riboprobes were transcribed *in vitro* from sense oligonucleotides containing a T7 promoter site (5'-GCCCTATAGTG AGTCGTATTAC-3') at their 3'-end using the MEGAshortscript T7 Transcription kit and digoxigenin-conjugated UTP (Roche, Indianapolis, IN). A mix of five non-overlapping riboprobes with matching GC content was used to detect each mRNA.

*Gfp*.1: 5'-GATGCCACCTACGGCAAGCTGACCCTGAA GTTCATCTGCACCACCGGCAAG-3'
*Gfp*.2: 5'-GACCACATGAAGCAGCACGACTTCTTCA AGTCCGCCATGCCCGAAGGCTAG-3'
*Gfp*.3: 5'-ACTTCAAGGAGGACGGCAACATCCTGGG GCACAAGCTGGAGTACAACTACG-3'
*Gfp*.4: 5'-AAGCAGAAGAACGGCATCAAGGTGAACTT CAAGATCCGCCACAACATCGAG-3'
*Gfp*.5: 5'-AGTTCGTGACCGCCGGGATCACTCTC GGCATGGACGAGCTGTACAAGG-3'
*Pafah1b1*.1: 5'-GCTCCGGTGGAATGAACCTTACTTGTTGA CTGGTTGCTGATTGGATTCAC-3'
*Pafah1b1*.2: 5'-GGATCTGAGACTAAAAAAAGTGGCAAGCCT GGACCCTTCTTGCTATCTGG-3'
*Pafah1b1*.3: 5'-CTGTTGGTGCCTGACTTGATGGCCTCATTT TGGGGAAAGTGGTTATTAGG-3'
*Pafah1b1*.4: 5'-CTAAGCTGAGAGAAAGTCACTTTATTCTCC CCTCTAATGGGCCATTCACC-3'
*Pafah1b1*.5: 5'-TACTGTTTTCTCTGTCTGCTGTCTAACCCTG TGCCTTGCCTGGGATAAGG-3'
*Dctn1*.1: 5'-TTGGAGATCCTCAAGGCTGAAATTGAAGAGA AAGGCTCTGATGGGGCTGC-3'
*Dctn1*.2: 5'-TCACCAAGGCCATCAAGTACTACCAGCA TCTGTACAGCATCCACCTCGCT-3'
*Dctn1*.3: 5'-CAACAGATATTGCTCTTCTTCTGCGAG ACCTGGAAACATCCTGCAGTGAC-3'
*Dctn1*.4: 5'-AAGGATGCTGATGAGCGAATCGAGAAAGT TCAGACTCGGCTGGAGGAGAC-3'
*Dctn1*.5: 5'-GGCCAAGGAAGAGCAGCAAGACGACACAGT CTACATGGGCAAAGTGACCT-3'
*Nude1*.1: 5'-ACAAGCCCCGGACACCCATGCCCAGCTCAG GGGAAACCAAAAGGACAGAC-3'
*Nude1*.2: 5'-TAGCATCATGCAAGAACTTCATGTATGAT CAGTCCCCAAGCCGGAAAAGC-3'
*Nude1*.3: 5'-TTGAGAAATAAGAGACCACATCACCAGCATT TACTTTGTGCTATGCAGCC-3'
*Nude1*.4: 5'-ATCTAGGACTGGGAGACTACAGAGGCAAAA GGCTATATTTAAACCATCTG-3'
*Nude1*.5: 5'-AAGCACGAGATGGAGATTACCACCAGCTGC TCCATCGGATGTGGAAAAAT-3'
*Nudel1*.1: 5'-AGCAGTTCGGGAACGGCAACAGGAAGTGACCCGAA AGTCTGCTCCCAGCT-3'
*Nudel1*.2: 5'-ATCCCGAAAATCCTATGTTCCAGGG AGCGTTAACTGTGGGGTAATGAACA-3'

*Nudel1.3:* 5′-CCGGCCTCCAGGTGGGGCCCCTGCCT
TCTTCCAGCAGCCCAGGACACTAC-3′

*Nudel1.4:* 5′-CGCCGTAGTGCCGTTGGTTTCACAT
GATTGCACTTTTGTGGGTCCCAAGT-3′

*Nudel1.5:* 5′-CACTGTCTGTACCTTCTGGCTTTATGT
AAGCAGTCCATTCCATTGCTTGT-3′.

FISH was performed as described previously[17]. Neurons grown in microfluidic chambers were fixed in 4% paraformaldehyde in PBS for 20 min at room temperature. Following three washes with PBS, the cells were permeabilized with 0.5% Triton X-100 in PBS and washed twice more with PBS. The coverslips were incubated with a total of 100 ng digoxigenin-labelled riboprobes (20 ng each of five distinct riboprobes) in 30 μl hybridization buffer (50% formamide, $2 \times$ SSC, 0.2% bovine serum albumin (BSA), 1 mg ml$^{-1}$ *E. coli* transfer RNA and 1 mg ml$^{-1}$ salmon sperm DNA) overnight at 37 °C. The coverslips were washed with constant agitation at 37 °C, first with 50% formamide in $2 \times$ SSC for 30 min followed by 50% formamide in $1 \times$ SSC for another 30 min. An additional three washes was done at room temperature with $1 \times$ SSC for 15 min each. The coverslips were washed three times with PBS containing 0.1% Tween-20 for 5 min each, blocked with 3% BSA in PBST for 30 min, and incubated with anti-digoxin (Sigma-Aldrich, DI-22; 1:500) and anti-β-III tubulin (Abcam, ab41489; 1:1,000) antibodies in blocking solution overnight at 4 °C. The coverslips were washed three times with PBST and incubated with fluorophore-conjugated Alexa secondary antibodies (1:2,000) for 1 h at room temperature, and washed and mounted with ProLong Gold antifade reagent. β-III tubulin staining was used to generate a mask within which the intensity of the FISH signal was quantified. Average fluorescence intensity of axonal fields that were incubated with a *Gfp* probe was subtracted from the fluorescence intensities resulting from hybridization with *Pafah1b1* or *Dctn1* riboprobes.

**Quantitative immunofluorescence imaging.** Axons of neurons grown in microfluidic chambers were exposed to 0, 5 or 100 ng ml$^{-1}$ NGF for 10 min, either in the presence of protein synthesis inhibitors or after pretreatment with siRNAs. Neurons were fixed with 4% paraformaldehyde/4% sucrose in PBS for 20 min at room temperature. The coverslips were washed three times in PBS, blocked for 1 h with BGT buffer (3% BSA, 0.25% Triton X-100 and 100 mM glycine) and incubated with primary antibodies against β-III tubulin (Abcam, ab7751; 1:500), 4EBP1, p-4EBP1 (Cell Signaling Technology, 1:1,000), puromycin (Millipore, MABE343; 1:250), Lis1 (Sigma-Aldrich, SAB3500302, 1:400) or p150$^{Glued}$ (Abcam, ab11806, 1:500). Images of distal axons or growth cones were acquired in Z-stacks using a Plan-Apochromat $\times 63/1.40$ oil objective.

**siRNA and LNA transfection.** Axon-specific silencing of *Pafah1b1* and *Dctn1* mRNAs was achieved by transfecting siRNAs into axons concomitant with the axonal medium change on DIV 3 using NeuroPORTER (Genlantis, San Diego, CA) as the transfection reagent. The following siRNAs were used to target rat *Pafah1b1* (NM_031763.3): 5′-CCUUUGACCACAGUGGCAAACUCUU-3′ and 5′-GGA UUUCCAUAAGACGGCACCCUAU-3′; and *Dctn1* (NM_024130.1): 5′-GAGCGC UCCUUAGAUUUCCUCAUCG-3′ and 5′-GACAUCCGUCAGUUCUGCAAGA AGA-3′. Negative control siRNA was purchased from ThermoFisher Scientific (Stealth RNAi siRNA Negative Control Med GC Duplex #3).

LNAs were transfected into the cell body compartment on DIV 3 using NeuroPORTER. The following high-performance liquid chromatography-purified LNAs (Exiqon, Woburn, MA) were used to target bases 1,878–1,891 (control LNA) or 1,905–1,921 (targeting the CUGU motif APC-binding site) of rat *Pafah1b1* mRNA: 5′-CA + TGAA + TACTT + TGT-3′ and 5′-A + TTTA + CAGTA + TACAA + TT-3′, respectively; preposed + signifies LNA base.

**Immunoblot.** The efficacy of each siRNA employed was validated by western blotting of endogenous protein from rat C6 glioma cells transfected at 50–80% confluence and cultured for 72 h to allow time for effective knockdown. Cells were lysed in RIPA buffer and proteins were resolved by 4–12% SDS–polyacrylamide gel electrophoresis on NuPAGE Bis-Tris gels, electrotransferred to Immobilon-P PVDF membranes (Millipore), blocked with 5% (w/v) nonfat milk, probed with primary antibodies followed by incubation with horseradish peroxidase-conjugated secondary antibodies (Pierce) and visualized with SuperSignal West Pico Chemiluminescent substrate (Pierce). Protein quantification was performed using ImageJ (NIH) software. The primary antibodies used for loading controls were: β-actin (1:10,000, Millipore) and cofilin (1:1,000, Cell Signaling Technology). horseradish peroxidase-conjugated secondary antibodies were used at 1:2,000. Images have been cropped for presentation. Full-size images are presented in Supplementary Fig. 4.

**TUNEL and calcein AM staining.** TUNEL was performed on fixed samples using the DeadEnd Fluorometric TUNEL System (Promega, Madison, WI), and nuclei were counterstained with 4,6-diamidino-2-phenylindole. Survival was analysed using calcein staining in living cells. Cell bodies were incubated with 4.17 μg ml$^{-1}$ calcein AM dye in dimethylsulphoxide for 40–60 min at 37 °C. Calcein was quenched with 15 mg ml$^{-1}$ bovine haemoglobin (Sigma-Aldrich), and nuclei were labelled with Hoechst stain. Cells were live imaged inside the microscope incubation chamber kept at 37 °C and 5% $CO_2$. TUNEL-positive nuclei and calcein-positive cells were scored in five fields per replicate that were proximal to the microgrooves.

**Real-time RT–PCR.** RNA was purified from the axonal compartments of microfluidic chambers using the PrepEase RNA Isolation kit (Affymetrix, Santa Clara, CA) and concentrated using the RNeasy MinElute Cleanup kit (Qiagen, Valencia, CA). A total amount of ∼2 ng was generally isolated from axonal lysates, which was concentrated into 10 μl for reverse transcription. Reverse transcription was performed using SuperScript III First-Strand Synthesis SuperMix for qRT–PCR. Real-time RT–PCR was performed with TaqMan Gene Expression master mix in a StepOnePlus Real-Time PCR instrument using the following conditions: an initial denaturation step at 95 °C for 10 min, followed by 40 cycles of denaturation at 95 °C for 15 s and extension at 60 °C for 1 min. *Pafah1b1* and *Dctn1* levels were normalized to *Gapdh*.

**RNA immunoprecipitation.** CUGU and control LNAs were transfected into dissociated DRGs and 24 h later the DRGs were lysed in RIP buffer (150 mM KCl, 25 mM Tris-HCl, 5 mM EDTA, 0.5 mM dithiothreitol, 0.5% NP40, + protease inhibitors). The cleared lysate was incubated with an APC antibody (Santa Cruz Biotechnology, sc-896; 1:500) overnight at 4 °C. Antibody–protein–RNA complexes were precipitate by incubation under agitation with Dynabeads for 1 h at 4 °C. The beads were washed five times in ice-cold RIP buffer. RNAzol RT was added to the beads, RNA was purified using the Direct-zol RNA MicroPrep kit (Zymo Research) with DNaseI treatment. Complementary was synthesized using the iScript Reverse Transcript Supermix for RT–qPCR. RT–PCR was run according to the guidelines for TaqMan Fast Advance Master Mix.

**Statistical analyses.** All experiments were performed in at least three biological replicates to gain sufficient power for meaningful statically analyses. Two means were compared by *t*-tests, whereas multiple means were compared using one-way ANOVAs with multiple comparisons testing. When comparing multiple groups in experiments with more than one variable, two-way ANOVA was performed. For all comparisons, normal distribution and variance were determined and appropriate statistical tests chosen.

**Data availability.** Data supporting the findings of this study are available within the article (and its Supplementary Information files) and from the corresponding author on reasonable request.

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

## Acknowledgements
This work was supported by the National Institute of Mental Health (R01MH096702 to U.H.), the National Institute of General Medical Sciences (F31GM116617 to J.C.M. and T32GM007367 for J.M.V. and J.C.M.) and the National Science Foundation Graduate Research Fellowship Program (to E.P.M.).

## Author contributions
J.M.V. with E.P.M. and J.C.M. performed and analysed all the experiments. U.H. conceived the project, and U.H. and J.M.V. designed the experiments and wrote the manuscript.

## Additional information

**Competing financial interests:** The authors declare no competing financial interests.

**Publisher's note**: 

