## [Peer Review File · Nature Communications]

Reviewers' comments:

Reviewer #1 (Remarks to the Author):

The Authors show that changing NGF levels either up or down causes a change (increase) in overall retrograde transport, and that this is associated with local translation of transcripts for Lis1 and P150 that are present in the axons.

This is original and of interest. Overall, the conclusions seem well supported, although the authors could provide better/additional movies/kymographs to document that transport really does change.

Although I support the work overall, there are a few things that are important to do before acceptance.

First, and most important, studies to date suggest that Lis1 typically works in conjunction with NudE and NudEL. In vitro studies of dynein with Lis1 alone suggest that dynein with lis1 only is blocked from advancing, and that it is only when the three-way dynein-NudE-Lis1 complex is functional that one has appropriate dynein force production as well as unimpaired motion. So, IT IS CRITICAL TO TEST FOR THE PRESENCE OF NUDE AND NUDEL TRANSCRIPTS, AND WHETHER THEY CHANGE IN RESPONSE TO NGF. This is the most significant criticism I have, and absolutely must be addressed.

Second (less important): How good is the locality of the effect? NGF is used to treat only the axon, but it could be affecting processes in the cell body as well. Is the NGF signal given to the axon changing protein expression in the cell body? The authors should use their 'protein expression assay' to look.

Third (minor): The authors should change the wording of " Together, these results indicate that the death signal, whose transport requires local Lis1 production, is likely active GSK3 β . " This doesn't follow at all, only that GSK3B is somewhere in the death signal pathway. This should be made explicit.

Reviewer #2 (Remarks to the Author):

This is a very confusing manuscript addressing local translation of dynein components in sensory axons. The study concurrently addresses changes in Lis1 and in dynactin in response to NGF stimulation and NGF withdrawal from the axon compartment of sensory neurons grown in culture in microfluidic chambers. There are undoubtedly a number of interesting findings in this paper. However, it is hard to sort out.

1. To assess trafficking of lysotracker positive vesicles, we are shown percentage of vesicles moving in each direction in 5 ng/ml. The next two graphs shows the "change in % lysotracker particles" which is stated to be "the percentage point difference to baseline condition are plotted". It would be much better to show the effects of NGF withdrawal and stimulation using the same method of assessment. This would make it much easier to follow and evaluate.

2. As evidence that the siRNAs are working, Figure S1 shows that the siRNAs (time not given) applied to rat glioma cells reduces the protein levels of these components. The next question that I would ask is are the siRNAs working in the DRG neurons, particularly in the axons. In 6E, the authors use Q-RT-PCR to analyze changes in axonal mRNAs in response to NGF withdrawal or stimulation for 12 hours. It appears that there is an increase in Pafah1b1 following NGF withdrawal, but no significant change in axonal mRNAs following NGF stimulation. Of note there is high variability in mRNA levels seen. Results

with siRNA are not shown for this experiment. When the same experiments are done by in situ hybridization, the graph in 6C does not provide statistical analysis of the signal for Pafah1b1 in NGF withdrawal versus baseline, so it is not clear if this result replicates by in situ hybridization. In these experiments, the siRNA for pafah1b1 appears to work only with NGF withdrawal or stimulation, and not at baseline. Similarly, the dctn1 siRNA only appears to work following stimulation, and not withdrawal or baseline. So how do we interpret figure 3- where there is no change in protein under baseline conditions, and only changes in dctn1 protein following stimulation? Can we conclude that this is about protein synthesis, or is about the uptake and efficacy of siRNA, or is about unmasking of mRNAs in axons (i.e. Buxbaum et al, Science, 2014), or does this reflect protein degradation rates?

3. Figure S2. Numerous studies have shown that survival of the cell bodies of sensory neurons can be maintained by NGF stimulation of cell bodies or axons. Figure S2 appears to show that anti-NGF in axons causes cell body apoptosis even when there is abundant NGF in the cell body compartment.

Reviewer #3 (Remarks to the Author):

In the study "Local synthesis of dynein cofactors matches retrograde transport to acutely changing demands", the authors use compartmentalized cultures of rat primary sensory neurons to study the role of locally translated minus-end motor co-factors in axons to regulate retrograde transport. Specifically, the authors focused on two transcripts encoding dynein regulators that have been identified in 3 independent studies as axonally localized mRNAs, Pafah1b1 encoding Lis1 and Dctn1 encoding p150glued. The authors use two induction paradigms, NGF withdrawal and stimulation and found that both lead to increased transport of LysoTracker-positive particles (late endosomes and autophagosomes) dependent on local translation. The study shows that NGF-induced transport of large vesicles required local synthesis of Lis1 while smaller NGF-signaling endosomes (consisting of NGF-TrkA complexes) required both dynein co-factors. Local translation of Lis1 was also triggered by NGF withdrawal and important for the transport of a death signal, which may be GSK3beta. In addition, NGF deprivation (but not stimulation) caused a significant increase in axonal pafah1b1 transcript levels. Finally, association of pafah1b1 mRNA with APC may be required for local translation in response to NGF stimulation but not for their axonal recruitment and translation upon NGF withdrawal. In summary, the authors claim that these studies demonstrate an important role for axonal protein synthesis of dynein regulators for the transport of specific vesicles and identify association with APC as a mechanism to establish functionally distinct pools of a single transcript species in axons.

This study is novel and is expected to be of high interest to the field and the wider readership of this journal. It describes the discovery of novel mechanisms for the regulation of induced changes to axonal transport, which depend on the local translation of motor protein co-factors. It also raises several important questions, e.g. how withdrawal of a ligand can be a stimulus for translation, considering the short time needed to induce translation.

Experiments are of high quality, use state of the arts methods, and include proper controls. The manuscript is very well written and mostly succeeds in explaining the fairly complex experimental approaches and results. In this reviewer's opinion there are only a few issues that should be addressed before publication as listed below.

- 1) There is no control experiment to demonstrate how local translation is inhibited (in Fig 1 & 2) upon treatment with anisomycin and emetine in the compartmentalized chamber. Puromycylation or AHA-labeling can be used to show the effect on global protein synthesis in soma and axons under the conditions used.
- 2) The results are quite complex, and concern 2 transcripts, 2 paradigms, various cargoes, protein

levels, and RNA levels. A schematic summary of the results would be very helpful for the readership. This can be in the form of a table or a schematic drawing/model.

3) The authors claim that "association with RNA-binding proteins as a mechanism to establish functionally distinct pools of a single transcript species in axons." It is not clear if this is really all the same transcript. Differences in the sequence e.g. in the 3'UTR due to alternative splicing would be one potential mechanism and should be discussed.

4) Surprisingly, neither the introduction nor the discussion mention that these dynein modifiers play a role in neurologic disease. While the focus of this study is on basic mechanisms in neurons, this seems at least as relevant as functional studies in yeast.

5) The microfluidic chambers should be described in more detail in the methods section, unless they are commercially available (Xona). For the sake of reproducibility, their manufacture and dimensions of the PDMS devices should be described.

6) The legend for Figure 5 "Pro-apoptotic GSK3 β signaling from NGF-deprived axons requires axonally synthesized Lis1" seems exaggerated compared to the more precise descriptions in the text and should be toned down accordingly (see Results section: "These results indicate that the death signal, whose transport requires local Lis1 production, is likely active GSK3 β .")

Reviewer #4 (Remarks to the Author):

This manuscript investigates the local synthesis of dynein cofactors and retrograde transport in response to NGF in DRG axons in vitro. The manuscript provides evidence that NGF stimulation can differentially regulate the translation of dynein cofactors, Lis1 and p150, according to its concentration, thus controlling retrograde transport. Axonal synthesis of Lis1, not p150, is also induced by NGF withdrawal and is involved in the retrograde transport of the apoptotic signaling. Finally, based on Preitner et al., the authors uncover two distinct mechanisms underlying the axonal translation of Lis1, APC-independent and APC-dependent, according to the absence or presence of NGF respectively. This leads the authors to conclude that different stimulus-specific mechanisms underlie the translation of the same transcript. In addition, they suggest that the precise spatial localization of APC at the plus end of microtubules and the APC-dependent translation of Lis1 provides an ideal site for the initiation of the retrograde transport.

This is a very interesting study that reports novel findings showing that intra-axonal protein synthesis finely controls retrograde transport and matches it to the levels of NGF stimulation. It is particularly interesting that axonally synthesized Lis1 has functionally distinct pools depending on whether its transcripts are APC-associated or not. The experiments are well designed and the results are compelling, for the most part. There are, however, some additional experiments/revisions that would strengthen the results and conclusions (see below). Overall, the findings should be of interest to a wide audience of scientists.

Major points

1. The authors conducted LNA experiments to interfere with the interaction between Pafah1b1 (mRNA) and APC using an LNA oligomer targeted to a putative APC-binding site (CUGU-motif) in the 3'UTR of Pafah1b1 mRNAs. They concluded from this experiment that "Association of Lis1 transcripts with the microtubule plus-end tracking protein APC was required for their translation in response to NGF stimulation but not for their axonal recruitment and translation upon NGF withdrawal." However, the authors do not provide any evidence that the LNA they used actually inhibits the interaction between the APC protein and the "putative" APC binding site.

The CUGU motif was reported in a previous CLIP study as one of three consensus sequence motifs

identified within APC binding regions (Preitner, N., et al. 2014). However, the 3'UTR sequence of Pafah1b1 mRNAs (Refseq mRNAs) harbors more than 20 of CUGU motifs, and it is not clear why the authors focused on the single putative APC binding site among these many CUGU motifs. Did the CLIP data show a single cluster at this site?

2. A major part of their conclusion comes from the experiments using siRNAs, but the knockdown effect of these siRNAs is not conclusive. In Figure 6C and D, they showed quantitative FISH results in axons for siRNA-target mRNAs following the siRNA treatment. However, in several conditions, the siRNAs did not show any significant decrease of the levels of FISH signals (baseline-Pafah1b1 siRNA, baseline Dctn1-siRNA and withdrawal-Dctn1 siRNA). Why did the siRNAs not work in these conditions?

In the methods section (and Fig S1), the authors mentioned that they used two different siRNAs for each target. Did the two siRNAs show consistent phenotypes? Their results would be strengthened if they could rule out the possibility of off-target effects by carrying out rescue experiments (for several important phenotypes).

3. The authors refer to the 5ng/ml NGF condition as 'baseline' condition and discuss the related data accordingly. Although the NGF concentration is low, it is still strictly 'stimulation'. Therefore, it would be more accurate to state that NGF-induced retrograde transport in axons is PS-dependent at a high concentration of NGF and PS-independent at a low concentration. Previous studies have similarly reported cue-concentration PS-dependence with other cues and shown that different cue concentrations activate different mechanisms/responses (e.g. Sema3A; Manns et al. 2012). Along similar lines, the description 'constitutive unstimulated' given in the text is not entirely correct, given the presence of NFG, although not high concentration.

4. The legend of Fig.4 states that cargo with a diameter $\geq 1 \mu\text{m}$ was chosen for analysis but reason for this size selection is not justified. Early and late endosomes, lysosome and autophagosomes all fall within this size limit. The authors should clearly explain the reason for the choice and specify the categories of cargo that are targeted in their analysis. In addition, in order to claim that Lis1 is required specifically for high-load cargo only, as previously shown by Yi et al. 2011, the authors should carry out in parallel the same analysis after filtering for diameter $< 1 \mu\text{m}$ too. Does Yi et al. 2011 really show that p150 is not involved in high-load retrograde transport as referenced by the author (Line 144)?

5. The text states 'Retrograde movement of NGF signaling endosomes from axons to the cell body is widely considered to be required for the survival of neurons dependent upon target-derived neurotrophic support.' (Line 162). This is somewhat misleading given the current state of the field. Indeed, besides the papers referenced here, there is also an opposite mechanism nicely shown by Mok et al. 2009. The authors should reference also this work and consider keeping the statement more general, e.g. 'Regulation of axonal retrograde transport in response to NGF signaling is widely considered to be required for the survival of neurons...'

Line 169: Similarly, why do the authors state 'contrary to expectation', given the Campenot and Barde papers? The authors should be more accurate in their statements and give more credit to previous discoveries. The authors cite the Campenot paper only later (line 181) whereas this work should be mentioned in the introduction and throughout the manuscript to give a more balanced and complete view of the current state of the field and how their own findings fit into it.

6. To prove the interaction between APC and the mRNA of interest the authors could carry out a band-shift assay (Control vs LNA against the putative APC-binding site).

Fig.7B: This experiment was carried out 12 h after different NGF treatments. How does this relate to the 10 min timescale of the NGF treatment/depletion experiments? The authors should provide a

statistical analysis not only among the different NGF treatments but also within the same treatment in order to state that there is a significant decrease of a fixed Lis1 mRNA amount after LNA transfection.

Minor points

1. The images in Fig. 2C-2D appear blurred, can higher resolution/quality images be provided? The anti-p150 Ab gives quite a high background. It would be valuable to provide western blot data to verify antibody specificity.
2. The title of Figure 5 "Pro-apoptotic GSK3 β signaling from NGF-deprived axons requires axonally synthesized Lis1" is not quite accurate given that there are no data showing a relationship between GSK3 β and Lis1 in this manuscript.
3. In P10, the authors suggest that "The increase in axonal Lis1 abundance in CUGU LNA transfected DRGs at 0 ng ml⁻¹ was indistinguishable from naïve and control LNA neurons, while in NGF-stimulated axons transfection with the CUGU LNA prevented the increase in Lis1 levels." But the statistical values to conclude this are missing in the Figure 7C.
4. Line 172: The author writes 'Similar results were obtained when NGF was maintained in the cell bodies'. However, in the presence of NGF 100 ng/ml, Lis1 siRNA induces an increase in the dead cells. Can the author clarify this outcome?

We would like to thank the reviewers for their careful critique of our manuscripts. We have address all of their points as follows:

Reviewer #1 (Remarks to the Author):

The Authors show that changing NGF levels either up or down causes a change (increase) in overall retrograde transport, and that this is associated with local translation of transcripts for Lis1 and P150 that are present in the axons.

This is original and of interest. Overall, the conclusions seem well supported, although the authors could provide better/additional movies/kymographs to document that transport really does change.

We are now providing higher resolution versions of all figures including larger images of the kymographs.

Although I support the work overall, there are a few things that are important to do before acceptance.

First, and most important, studies to date suggest that Lis1 typically works in conjunction with NudE and NudEL. In vitro studies of dynein with Lis1 alone suggest that dynein with lis1 only is blocked from advancing, and that it is only when the three-way dynein-NudE-Lis1 complex is functional that one has appropriate dynein force production as well as unimpaired motion. So, IT IS CRITICAL TO TEST FOR THE PRESENCE OF NUDE AND NUDEL TRANSCRIPTS, AND WHETHER THEY CHANGE IN RESPONSE TO NGF. This is the most significant criticism I have, and absolutely must be addressed.

We agree with the reviewer, that Lis1 typically works in conjunction with NuDE and NudEL, however, we would like to point out that this fact does not necessitate that NudE and NudEL are locally synthesized together with Lis1. There are examples in the literature that only a single protein of a larger complex is locally translated with the rest of the complex being provided by synthesis in the cell body and anterograde transport (e.g. Hengst et al., Nat Cell Biol, 2009). We have performed the requested experiment. Both NudE and NudEL are detectable by FISH in distal axons and their levels are significantly increased in NGF starved axons but not in NGF stimulated axons (new Supplementary Figure 2). We discuss the possibility that they are co-translated with Lis1 in the manuscript.

Second (less important): How good is the locality of the effect? NGF is used to treat only the axon, but it could be affecting processes in the cell body as well. Is the NGF signal given to the axon changing protein expression in the cell body? The authors should use their 'protein expression assay' to look.

The data requested by the reviewer are part of Figure 3a, b: in naïve (i.e. not siRNA transfected) neurons the expression levels of Lis1 or p150^{Glued} are not changed in cell bodies upon NGF addition to the axons.

Third (minor): The authors should change the wording of " Together, these results indicate that the death signal, whose transport requires local Lis1 production, is likely active GSK3 β . " This doesn't follow at all, only that GSK3B is somewhere in the death signal pathway. This should be made explicit.

We have changed the wording: '~~...is likely~~ involves active GSK3 β .'

Reviewer #2 (Remarks to the Author):

This is a very confusing manuscript addressing local translation of dynein components in sensory axons. The study concurrently addresses changes in Lis1 and in dynactin in response to NGF stimulation and NGF withdrawal from the axon compartment of sensory neurons grown in culture in microfluidic chambers. There are undoubtedly a number of interesting findings in this paper. However, it is hard to sort out.

We have added a model of the proposed mechanism as a new Supplementary Figure 3.

1. To assess trafficking of lysotracker positive vesicles, we are shown percentage of vesicles moving in each direction in 5 ng/ml. The next two graphs shows the "change in % lysotracker particles" which is stated to be "the percentage point difference to baseline condition are plotted". It would be much better to show the effects of NGF withdrawal and stimulation using the same method of assessment. This would make it much easier to follow and evaluate.

We have changed the graph style in Figures 1, 4 & 7 accordingly.

2. As evidence that the siRNAs are working, Figure S1 shows that the siRNAs (time not given) applied to rat glioma cells reduces the protein levels of these components.

We have now added the time (72 hours) to the legend of Supplementary Figure 1.

The next question that I would ask is are the siRNAs working in the DRG neurons, particularly in the axons.

We demonstrate that the siRNAs work in DRGs by western blotting in Supplementary Figure S1. Their knockdown effect in axons is demonstrated by FISH in Figure 6c, d.

In 6E, the authors use Q-RT-PCR to analyze changes in axonal mRNAs in response to NGF withdrawal or stimulation for 12 hours. It appears that there is an increase in Pafah1b1 following NGF withdrawal, but no significant change in axonal mRNAs following NGF stimulation. Of note there is high variability in mRNA levels seen. Results with siRNA are not shown for this experiment. When the same experiments are done by in situ hybridization, the graph in 6C does not provide statistical analysis of the signal for Pafah1b1 in NGF withdrawal versus baseline, so it is not clear if this result replicates by in situ hybridization.

The effect replicates by FISH and statistical significance is indicated by an asterisk over the *Pafah1b1* bar in the NGF withdrawal condition.

In these experiments, the siRNA for pafaph1b1 appears to work only with NGF

withdrawal or stimulation, and not at baseline. Similarly, the *dctn1* siRNA only appears to work following stimulation, and not withdrawal or baseline. So how do we interpret figure 3- where there is no change in protein under baseline conditions, and only changes in *dctn1* protein following stimulation? Can we conclude that this is about protein synthesis, or is about the uptake and efficacy of siRNA, or is about unmasking of mRNAs in axons (i.e. Buxbaum et al, Science, 2014), or does this reflect protein degradation rates?

siRNAs in axons normally do not knockdown silent mRNAs. The reviewer is correct in referring to Buxbaum et al. Silenced mRNAs are too tightly packaged to be accessible for the siRNA machinery. We included a discussion of this possibility in the Results section for Figure 6.

3. Figure S2. Numerous studies have shown that survival of the cell bodies of sensory neurons can be maintained by NGF stimulation of cell bodies or axons. Figure S2 appears to show that anti-NGF in axons causes cell body apoptosis even when there is abundant NGF in the cell body compartment.

We have removed Supplementary Figure 2, primarily because the survival signaling pathways triggered by NGF differ between locally and axonally applied NGF (e.g. Lonze, B.E., Riccio, A., Cohen, S., Ginty, D.D. (2002) Apoptosis, axonal growth defects, and degeneration of peripheral neurons in mice lacking CREB. Neuron 34, 371-385). Thus the results from the previous Figure S2 are difficult to compare with the remainder of the manuscript.

Reviewer #3 (Remarks to the Author):

In the study “Local synthesis of dynein cofactors matches retrograde transport to acutely changing demands”, the authors use compartmentalized cultures of rat primary sensory neurons to study the role of locally translated minus-end motor co-factors in axons to regulate retrograde transport. Specifically, the authors focused on two transcripts encoding dynein regulators that have been identified in 3 independent studies as axonally localized mRNAs, *Pafah1b1* encoding *Lis1* and *Dctn1* encoding *p150glued*. The authors use two induction paradigms, NGF withdrawal and stimulation and found that both lead to increased transport of LysoTracker-positive particles (late endosomes and autophagosomes) dependent on local translation. The study shows that NGF-induced transport of large vesicles required local synthesis of *Lis1* while smaller NGF-signaling endosomes (consisting of NGF-TrkA complexes) required both dynein co-factors. Local translation of *Lis1* was also triggered by NGF withdrawal and important for the transport of a death signal, which may be GSK3beta. In addition, NGF deprivation (but not stimulation) caused a significant increase in axonal *pafah1b1* transcript levels. Finally, association of *pafah1b1* mRNA with APC may be required for local translation in response to NGF stimulation but not for their axonal recruitment and translation upon NGF withdrawal. In summary, the authors claim that these studies demonstrate an important role for axonal protein synthesis of dynein regulators for the transport of specific vesicles and identify association with APC as a mechanism to

establish functionally distinct pools of a single transcript species in axons.

This study is novel and is expected to be of high interest to the field and the wider readership of this journal. It describes the discovery of novel mechanisms for the regulation of induced changes to axonal transport, which depend on the local translation of motor protein co-factors. It also raises several important questions, e.g. how withdrawal of a ligand can be a stimulus for translation, considering the short time needed to induce translation.

Experiments are of high quality, use state of the arts methods, and include proper controls. The manuscript is very well written and mostly succeeds in explaining the fairly complex experimental approaches and results. In this reviewer's opinion there are only a few issues that should be addressed before publication as listed below.

1) There is no control experiment to demonstrate how local translation is inhibited (in Fig 1 & 2) upon treatment with anisomycin and emetine in the compartmentalized chamber. Puromycylation or AHA-labeling can be used to show the effect on global protein synthesis in soma and axons under the conditions used.

We have now added the data from the proposed puromycylation assay: anisomycin and emetine inhibit puromycylation in response to NGF stimulation and withdrawal in axons but do not change puromycylation in the cell bodies (new Figure 1f, g).

2) The results are quite complex, and concern 2 transcripts, 2 paradigms, various cargoes, protein levels, and RNA levels. A schematic summary of the results would be very helpful for the readership. This can be in the form of a table or a schematic drawing/model.

We have added a schematic as new Supplementary Figure 3.

3) The authors claim that "association with RNA-binding proteins as a mechanism to establish functionally distinct pools of a single transcript species in axons." It is not clear if this is really all the same transcript. Differences in the sequence e.g. in the 3'UTR due to alternative splicing would be one potential mechanism and should be discussed.

The reviewer brings up an interesting possibility. However, at least in the case for Pafah1b1 this is unlikely as there are no splice variants of Lis1 with an alternative last exon and additionally the CUGU motif is too close to the STOP codon to be excluded by alternative polyadenylation site usage.

4) Surprisingly, neither the introduction nor the discussion mention that these dynein modifiers play a role in neurologic disease. While the focus of this study is on basic mechanisms in neurons, this seems at least as relevant as functional studies in yeast.

We have added a discussion of the potential implications of our study for neurological disorders.

5) The microfluidic chambers should be described in more detail in the methods section, unless they are commercially available (Xona). For the sake of reproducibility, their manufacture and dimensions of the PDMS devices should be described.

We have added detail and dimensions of the PDMS devices in the Methods section; they were not purchased from Xona).

6) The legend for Figure 5 “Pro-apoptotic GSK3 β signaling from NGF-deprived axons requires axonally synthesized Lis1” seems exaggerated compared to the more precise descriptions in the text and should be toned down accordingly (see Results section: “These results indicate that the death signal, whose transport requires local Lis1 production, is likely active GSK3 β .”)

We changed the figure title to: *Figure 5. Pro-apoptotic signaling from NGF-deprived axons requires axonally synthesized Lis1 and active GSK3 β*

Reviewer #4 (Remarks to the Author):

This manuscript investigates the local synthesis of dynein cofactors and retrograde transport in response to NGF in DRG axons in vitro. The manuscript provides evidence that NGF stimulation can differentially regulate the translation of dynein cofactors, Lis1 and p150, according to its concentration, thus controlling retrograde transport. Axonal synthesis of Lis1, not p150, is also induced by NGF withdrawal and is involved in the retrograde transport of the apoptotic signaling. Finally, based on Preitner et al., the authors uncover two distinct mechanisms underlying the axonal translation of Lis1, APC-independent and APC-dependent, according to the absence or presence of NGF respectively. This leads the authors to conclude that different stimulus-specific mechanisms underlie the translation of the same transcript. In addition, they suggest that the precise spatial localization of APC at the plus end of microtubules and the APC-dependent translation of Lis1 provides an ideal site for the initiation of the retrograde transport.

This is a very interesting study that reports novel findings showing that intra-axonal protein synthesis finely controls retrograde transport and matches it to the levels of NGF stimulation. It is particularly interesting that axonally synthesized Lis1 has functionally distinct pools depending on whether its transcripts are APC-associated or not. The experiments are well designed and the results are compelling, for the most part. There are, however, some additional experiments/revisions that would strengthen the results and conclusions (see below). Overall, the findings should be of interest to a wide audience of scientists.

Major points

1. The authors conducted LNA experiments to interfere with the interaction between Pafah1b1 (mRNA) and APC using an LNA oligomer targeted to a putative APC-binding site (CUGU-motif) in the 3'UTR of Pafah1b1 mRNAs. They concluded from this experiment that “Association of Lis1 transcripts with the microtubule plus-end tracking protein APC was required for their translation in response to NGF stimulation but not for their axonal recruitment and translation upon NGF withdrawal.” However, the authors do

not provide any evidence that the LNA they used actually inhibits the interaction between the APC protein and the “putative” APC binding site.

We have performed RNA-immunoprecipitation experiments with an antibody against APC. The results are shown in new Figure 7b: *Pafah1b1* immunoprecipitates with APC and this interaction is disrupted by the CUGU LNA.

The CUGU motif was reported in a previous CLIP study as one of three consensus sequence motifs identified within APC binding regions (Preitner, N., et al. 2014). However, the 3'UTR sequence of *Pafah1b1* mRNAs (Refseq mRNAs) harbors more than 20 of CUGU motifs, and it is not clear why the authors focused on the single putative APC binding site among these many CUGU motifs. Did the CLIP data show a single cluster at this site?

We have added details about the identification of the CUGU motif (based on CLIP data by Preitner et al.).

2. A major part of their conclusion comes from the experiments using siRNAs, but the knockdown effect of these siRNAs is not conclusive. In Figure 6C and D, they showed quantitative FISH results in axons for siRNA-target mRNAs following the siRNA treatment. However, in several conditions, the siRNAs did not show any significant decrease of the levels of FISH signals (baseline-*Pafah1b1* siRNA, baseline *Dctn1*-siRNA and withdrawal-*Dctn1* siRNA). Why did the siRNAs not work in these conditions?

The reviewer is correct; siRNAs in axons target predominantly active translated mRNAs. Silenced mRNAs in axons are tightly packed with RNP, making them inaccessible for the siRNA (see also Reviewer 1, point 2).

In the methods section (and Fig S1), the authors mentioned that they used two different siRNAs for each target. Did the two siRNAs show consistent phenotypes? Their results would be strengthened if they could rule out the possibility of off-target effects by carrying out rescue experiments (for several important phenotypes).

Rescue experiments in axons are technically virtually impossible to conduct in a controlled manner: It is not enough to express an siRNA resistant mRNA in the neurons, it also has to contain the axonal localization sequence and other regulatory elements. Thus, the resistant mRNA would have to contain the endogenous 5' and 3'UTRs. Complicating the matter, overexpression of such a construct would act as a sponge for RNA-binding proteins causing a multitude of side effects. On the other hand, too low expression levels will fail to rescue. In practice, it is nearly impossible to express the rescue constructs at just the right level to have axonal localization and no sponge-effect.

3. The authors refer to the 5ng/ml NGF condition as ‘baseline’ condition and discuss the related data accordingly. Although the NGF concentration is low, it is still strictly ‘stimulation’. Therefore, it would be more accurate to state that NGF-induced retrograde transport in axons is PS-dependent at a high concentration of NGF and PS-independent at a low concentration. Previous studies have similarly reported cue-concentration PS-dependence with other cues and shown that different cue concentrations activate different mechanisms/responses (e.g. *Sema3A*; Manns et al. 2012). Along similar lines,

the description 'constitutive unstimulated' given in the text is not entirely correct, given the presence of NFG, although not high concentration.

We have amended the text to clarify baseline refers to the PS-independence. We state that even at this low NGF concentration there is still NGF signaling (or stimulation) such as survival signaling.

4. The legend of Fig.4 states that cargo with a diameter $\geq 1 \mu\text{m}$ was chosen for analysis but reason for this size selection is not justified. Early and late endosomes, lysosome and autophagosomes all fall within this size limit. The authors should clearly explain the reason for the choice and specify the categories of cargo that are targeted in their analysis.

The different axonal vesicles that are labelled by LysoTracker (late endosomes and autophagosomes, see Maday et al., J Cell Biol, 2012) are listed in the text. We focused on larger LysoTracker positive particles because finer sizes make it difficult to identify and follow individual particles (see Yi et al., J Cell Biol, 2011). We have included this information in the Methods section.

In addition, in order to claim that Lis1 is required specifically for high-load cargo only, as previously shown by Yi et al. 2011, the authors should carry out in parallel the same analysis after filtering for diameter $< 1 \mu\text{m}$ too.

We believe that this is a misunderstanding. We do not claim that Lis1 is specifically required for high-load only, in fact we found that it is required for the retrograde transport of NGF signaling endosomes (Figure 4d) which are smaller than $1 \mu\text{m}$.

Does Yi et al. 2011 really show that p150 is not involved in high-load retrograde transport as referenced by the author (Line 144)?

The reviewer is correct; we have corrected the sentence to accurately reflect the finding in Yi et al..

5. The text states 'Retrograde movement of NGF signaling endosomes from axons to the cell body is widely considered to be required for the survival of neurons dependent upon target-derived neurotrophic support.' (Line 162). This is somewhat misleading given the current state of the field. Indeed, besides the papers referenced here, there is also an opposite mechanism nicely shown by Mok et al. 2009. The authors should reference also this work and consider keeping the statement more general, e.g. 'Regulation of axonal retrograde transport in response to NGF signaling is widely considered to be required for the survival of neurons...'

We have changed the beginning of this paragraph accordingly and introduce the Campenot study at this point already.

Line 169: Similarly, why do the authors state 'contrary to expectation', given the Campenot and Barde papers? The authors should be more accurate in their statements and give more credit to previous discoveries. The authors cite the Campenot paper only later (line 181) whereas this work should be mentioned in the introduction and throughout the manuscript to give a more balanced and complete view of the current

state of the field and how their own findings fit into it.

We changed to wording to emphasize that the results are contrary to the signaling endosome hypothesis but in line with the Campenot model.

6. To prove the interaction between APC and the mRNA of interest the authors could carry out a band-shift assay (Control vs LNA against the putative APC-binding site).

We have added results from an RNA immunoprecipitation assay in new Figure 7b. CUGU LNA interferes with *Pafah1b1*-APC association.

Fig.7B: This experiment was carried out 12 h after different NGF treatments. How does this relate to the 10 min timescale of the NGF treatment/depletion experiments?

This experiment (now Figure 7c) looks at the recruitment of mRNA into axons in the presence of LNA that were transfected into the cell bodies, a process that takes several hours. It is comparable to Figures 6 c, d, in which the same time course is used. The 10 min time course is used to study axonal processes.

The authors should provide a statistical analysis not only among the different NGF treatments but also within the same treatment in order to state that there is a significant decrease of a fixed Lis1 mRNA amount after LNA transfection.

To address this question we have performed a two-way ANOVA analysis and given the results as p-values in the main text. The control LNA had no discernible effect compared to naïve axons (two-way ANOVA $p=0.7585$), while transfection with the CUGU LNA caused reduced *Pafah1b1* levels in all three conditions ($p=0.008$).

Minor points

1. The images in Fig. 2C-2D appear blurred, can higher resolution/quality images be provided? The anti-p150 Ab gives quite a high background. It would be valuable to provide western blot data to verify antibody specificity.

The perceived blurriness of the figures might have been caused by us embedding it into the manuscript file. We provide now high resolution images. The requested verification of antibody specificity by western blot is provided for both the Lis1 and p150^{Glued} antibody in Supplementary Figure 1.

2. The title of Figure 5 “Pro-apoptotic GSK3 β signaling from NGF-deprived axons requires axonally synthesized Lis1” is not quite accurate given that there are no data showing a relationship between GSK3 β and Lis1 in this manuscript.

We have changed see figure title (see Reviewer 3, point 6).

3. In P10, the authors suggest that “The increase in axonal Lis1 abundance in CUGU LNA transfected DRGs at 0 ng ml⁻¹ was indistinguishable from naïve and control LNA neurons, while in NGF-stimulated axons transfection with the CUGU LNA prevented the increase in Lis1 levels.” But the statistical values to conclude this are missing in the Figure 7C.

We have added the statistical information to the figure.

4. Line 172: The author writes 'Similar results were obtained when NGF was maintained in the cell bodies'. However, in the presence of NGF 100 ng/ml, Lis1 siRNA induces an increase in the dead cells. Can the author clarify this outcome?

We have removed Supplementary Figure 2. See also response to Reviewer 2, point 3.

Reviewers' comments:

Reviewer #1 (Remarks to the Author):

I now support publication. My Concerns were addressed. Might want to cite new work out (Load-induced enhancement of Dynein force production by LIS1–NudE in vivo and in vitro) in Nature Communications, since it supports the in vivo function of NudE/Lis1 to improve force production, but this is only a mild suggestion, and certainly not necessary for publication.

Reviewer #2 (Remarks to the Author):

This is a revised manuscript addressing translation of dynein components in axons in response to NGF stimulation or withdrawal.

One major question that remains regards the “control conditions”. This is a very technical but important issue. The controls used are cultures that remain in 5 ng/ml NGF. Did the investigators exchange the media in these cultures, or were they left untouched? If the latter, this may explain some of the shared changes seen with deprivation and stimulation, as mechanical stimulation clearly alters these neurons.

It is interesting that siRNAs apparently only work in the stimulated or unstimulated conditions (see figure 6), and are not effective under baseline conditions. Could the authors provide a reference for this phenomenon? Buxbaum et al demonstrate that masked mRNAs exist, but does not address their accessibility to siRNA. Also, it would be nice to show effects on protein as well as on mRNA.

The authors postulate two pools of RNAs for Lis1/Pafah1b1. Are they suggesting that these are anatomically distinct? Can they see distinct pools in the axons?

Greater discussion of statistical testing seems indicated in several figures.

Reviewer #3 (Remarks to the Author):

This reviewer is satisfied with the author's response. All concerns have been addressed adequately, and the changes to the manuscript have significantly improved the clarity and quality of this interesting study.

Reviewer #4 (Remarks to the Author):

The authors have addressed almost all the points raised in my original review, with one minor omission. They did not directly answer the point - "2. ...In the methods section (and Fig S1), the authors mentioned that they used two different siRNAs for each target. Did the two siRNAs show consistent phenotypes?" The thought here was simply that if the two siRNAs gave the same result, then this would further strengthen their results. The author's answer gave a valid explanation about how it was not technically feasible to conduct a rescue experiment in axons. However, it would be valuable to include information in the manuscript about whether the two different siRNAs they used gave the same phenotype. Overall, this is an exciting study and should be of interest to a broad audience of neuroscientists.

We have addressed the last remaining points as follows:

Reviewer #1 (Remarks to the Author):

I now support publication. My Concerns were addressed. Might want to cite new work out (Load-induced enhancement of Dynein force production by LIS1–NudE in vivo and in vitro) in Nature Communications, since it supports the in vivo function of NudE/Lis1 to improve force production, but this is only a mild suggestion, and certainly not necessary for publication.

We have included this reference (p. 2).

Reviewer #2 (Remarks to the Author):

This is a revised manuscript addressing translation of dynein components in axons in response to NGF stimulation or withdrawal.

One major question that remains regards the “control conditions”. This is a very technical but important issue. The controls used are cultures that remain in 5 ng/ml NGF. Did the investigators exchange the media in these cultures, or were they left untouched? If the latter, this may explain some of the shared changes seen with deprivation and stimulation, as mechanical stimulation clearly alters these neurons.

In the control condition, the 5 ng ml⁻¹ media was exchanged for the very reason that the reviewer proposes. We have now changed the legend for Figure 1 to clarify this point (p. 29).

It is interesting that siRNAs apparently only work in the stimulated or unstimulated conditions (see figure 6), and are not effective under baseline conditions. Could the authors provide a reference for this phenomenon? Buxbaum et al demonstrate that masked mRNAs exist, but does not address their accessibility to siRNA. Also, it would be nice to show effects on protein as well as on mRNA.

We are now citing a previous publication (Baleriola et al. Cell 2014), in which we observed the same refractory response of silenced mRNAs to local siRNA transfection (p. 9). The effect of axonal siRNA transfections on protein levels are part of Figure 3.

The authors postulate two pools of RNAs for Lis1/Pafah1b1. Are they suggesting that these are anatomically distinct? Can they see distinct pools in the axons?

Although we suggest that there are two different pools of Lis1 transcripts, this is only suggesting that two pools are present, but not commenting on specific features of their organization. At this time we do not have any insight into how to visualize the proposed to different pools.

Greater discussion of statistical testing seems indicated in several figures.

The statistical tests used are given in each figure legend, together with the n, the number of replicates, information of whether these are biological or technical replicates, and the p

values.

Reviewer #3 (Remarks to the Author):

This reviewer is satisfied with the author's response. All concerns have been addressed adequately, and the changes to the manuscript have significantly improved the clarity and quality of this interesting study.

Reviewer #4 (Remarks to the Author):

The authors have addressed almost all the points raised in my original review, with one minor omission. They did not directly answer the point - "2. ...In the methods section (and Fig S1), the authors mentioned that they used two different siRNAs for each target. Did the two siRNAs show consistent phenotypes?" The thought here was simply that if the two siRNAs gave the same result, then this would further strengthen their results. The author's answer gave a valid explanation about how it was not technically feasible to conduct a rescue experiment in axons. However, it would be valuable to include information in the manuscript about whether the two different siRNAs they used gave the same phenotype.

We have now included the information that both siRNAs showed consistent phenotypes (p. 5).

Overall, this is an exciting study and should be of interest to a broad audience of neuroscientists.